# Economically optimal safety targets for interdependent flood defences in a graph-based approach with an efficient evaluation of expected annual damage estimates

Egidius Johanna Cassianus Dupuits[1], Ferdinand Lennaert Machiel Diermanse[2], and Matthijs Kok[1]

[1]Delft University of Technology, Faculty of Civil Engineering and Geosciences, P.O. Box 5048, 2600 GA Delft, Netherlands
[2]Deltares Unit Inland Water Systems, department of Flood Risk Management, P.O. Box 177, 2600 MH Delft, Netherlands

*Correspondence to:* E.J.C. Dupuits (e.j.c.dupuits@tudelft.nl)

**Abstract.** Flood defence systems can be seen as multiple interdependent flood defences. This paper advances an approach for finding an optimal configuration for flood defence systems, based on an economic cost-benefit analysis with an arbitrary number of interdependent flood defences. The proposed approach is based on a graph algorithm and is, thanks to some beneficial properties of the application, able to represent large graphs with strongly reduced memory requirements. Furthermore, computational efficiency is achieved by delaying cost calculations until they are actually needed by the graph algorithm. This significantly reduces the required number of computationally expensive flood risk calculations. In this paper, we conduct a number of case studies to compare the optimal paths found by the proposed approach with the results of competing methods that generate identical results. The proposed approach is set up in a generic way and implements the shortest-path approach for optimising cost-benefit analyses of interdependent flood defences with computationally expensive flood risk calculations.

## 1 Introduction

Concerns regarding the safety of people and assets in flood prone areas has led to the construction of flood defence systems all around the world. Some flood prone areas, for example a large part of the Netherlands, face huge potential loss of life and economic value in case heavy flooding occurs. This has led to extensive research regarding the estimation of flood risk in flood prone areas. Coupled to this quantification of the flood risk, is the question of 'how safe' a flood prone area should be and what the acceptable risk should be (Vrijling et al., 1998). An often used approach to help answer this question is a cost-benefit analysis.

Economic optimisation of flood defences, as applied in the Netherlands, is based on a cost-benefit analysis of the sum of the annual flood risks balanced against the sum of the investment costs for flood defences. This type of cost-benefit analysis was originally developed in the 1950's by Van Dantzig (1956) and is still used and discussed to this day (Eijgenraam, 2006; Kind, 2014). The basic principle behind the economic optimisation of flood defences is finding the minimum of the total costs as illustrated in Figure 1. The total costs ($TC$, Eq. 1) are the sum of the annual risk costs ($\sum_{t=0}^{p} R(t)$) and investment costs ($\sum_{t=0}^{p} I(t)$) over a given time period ($p$ years). The total costs are expressed as the present value of the (future) annual risk costs and investment costs, which means these costs are discounted at a discount rate $r$. The annual risk cost $R(t)$ is defined

in Eq. 2 as the annual probability of flooding at time $t$ ($P_{flood,t}$), multiplied by the expected damages due to flooding at time $t$ ($D_{flood,t}$). An alternative term for the annual risk cost is the Expected Annual Damage, or EAD. Generally speaking, a larger investment will lead to a lower EAD; this is where the economic optimisation tries to find an optimal solution (i.e. the lowest total cost).

$$TC = \sum_{t=0}^{p} R(t)e^{-rt} + \sum_{t=0}^{p} I(t)e^{-rt} \qquad (1)$$

$$R(t) = \text{EAD}(t) = P_{flood,t} \cdot D_{flood,t} \qquad (2)$$

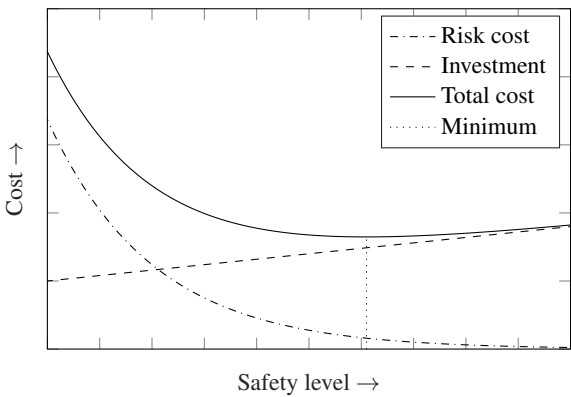

**Figure 1.** Schematic view of an economic cost-benefit analysis for a flood defence. The total costs are the sum of the risk and investment costs, and the optimum can be found at the minimum of the total costs.

Recent publications regarding economically optimal safety targets for the Netherlands can be found in the publications by Eijgenraam (2006), Brekelmans et al. (2012) and Zwaneveld and Verweij (2014b, a). In Eijgenraam (2006) and Eijgenraam et al. (2016), a set of equations were derived which describe the economically optimal safety target for a single homogeneous flood defence system (i.e. dike ring). Because they incorporated influence of time-dependent parameters such as economic growth and climate model parameters, these equations also describe the number of (repeated) investments, as well as the optimal time between these investments. Repeated investments are necessary to 'repair' the effect of, for example, economic growth (i.e. a higher expected losses in case of a flood) or subsidence (i.e. a higher flood probability). A schematic view of the result of such an economic optimisation with time-dependent parameters is shown in Figure 2. This figure shows that as the safety level goes down over time, recurring investments are needed to repair the effects over time of time-dependent parameters such as economic growth and climate change.

The equations described in Eijgenraam et al. (2016) are analytically solvable and the method results in a global minimum of the total costs for a relatively simple homogeneous system. However, dike rings in the Netherlands often consist of mutually different, non-homogeneous sections in which case, the homogenous case needs to be extended to account for these non-

homogeneous sections. In Brekelmans et al. (2012), a possible, heuristic solution is given by modelling the problem as a mixed-integer nonlinear programming (MINLP) problem. Zwaneveld and Verweij (2014b) improved on this method by developing a graph-based modelling approach to solve the non-homogeneous case to proven optimality.

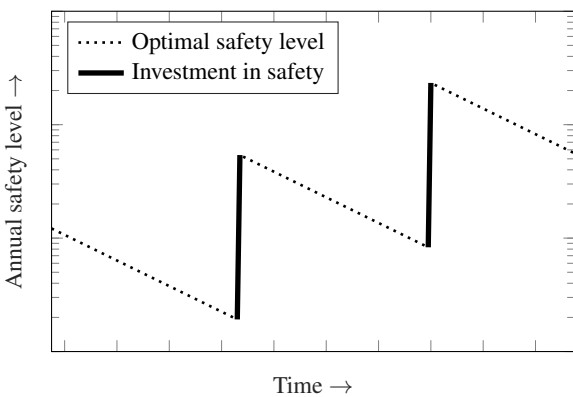

**Figure 2.** Schematic view of an economic cost-benefit analysis for a flood defence, with time-dependent parameters. Because of these time-dependent parameters (e.g. economic growth or subsidence), recurring investments in safety are needed.

Eijgenraam (2006), Brekelmans et al. (2012) and Zwaneveld and Verweij (2014b) assess independent flood prone areas in
which individual flood defences within a dike ring area fail under identical circumstances. No interdependencies exists in their modelling approaches, however, the notion of interdependent flood defences expresses that failure of one flood defence might alter the EAD of other defences. Moreover, a dike ring may fail under different circumstances. A practical example of a flood defence system with multiple interdependent flood defences is shown in Figure 3. In this figure, a breach occurring at upstream area B impacts the flood risk at area A. This impact can either increase or decrease the flood risk at area A. An increase would
occur if a shortcut is formed between area A and an already flooded area B at arrow 2. Whereas a decrease would occur if the breach at arrow 1 reduces the probability of a breach at arrow 3 (because part of the river discharge is diverted into area B). This notion of interdependent flood defences has been, from a flood risk perspective, the main topic of a number of recent papers (e.g. Vorogushyn et al. (2010, 2012); Courage et al. (2013); De Bruijn et al. (2014)). All of these papers showed that viewing the flood defence system as a whole, will result in different EAD estimates than viewing the flood defences as separate,
independent defences.

As the EAD changes, the economic optimisation will also be affected. Thus, it makes sense to explicitly integrate the effect of multiple interdependent flood defences on the EAD in the economic optimisation routines. A method to provide a modelling approach to the economic optimisation of a flood defence system with multiple dependent and independent dikes was first presented in Zwaneveld and Verweij (2014a). In their study (in Dutch), a graph-based modelling approach is used to obtain
economically optimal safety norms and heights for multiple lines of flood defences. Furthermore, they mentioned that the economic optimisation problem can be formulated in the form of a minimal cost flow graph or a shortest path problem. Three approaches (to solve economic optimal safety problems for multiple flood defences) were identified by Zwaneveld and Verweij

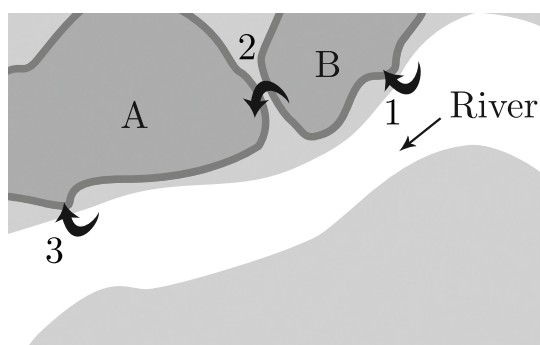

**Figure 3.** A hypothetical example of a system with interdependent flood defences. The flood risk in area A is is not only impacted by the flooding probability at its own defence (arrow 3), but also what happens at the flood defence of area B (arrow 1) and the connecting flood defences between area A and B (arrow 2).

(2014a) & Verweij (2014): (1) a heuristic approach based on closed form formulas, (2) a dynamic programming/shortest-path approach (as also used in Eijgenraam et al. (2016)) and (3) a branch-and-cut/ILP approach. In Zwaneveld and Verweij (2014a) the branch-and-cut/ILP-approach is preferred and applied. An English description of the model in Zwaneveld and Verweij (2014a) can be found in (Yüceoglu, 2015, Chapter 5).

However, a consequence of using an ILP approach in Zwaneveld and Verweij (2014a) is that, prior to starting the optimisation routine, all EAD estimates for each and every possible combination of flood defences in time need to be computed. Generally speaking, finding EAD estimates for a number of these combinations is not necessary. For example, it is unlikely that is economically optimal to keep all flood defences at their lowest level for the next 300 years. Calculating these EAD estimates can be costly, especially if hydrodynamic interactions are included since acquiring a single EAD estimate can take hours (De

Bruijn et al., 2014) or even days (Courage et al., 2013). In these cases, computational efficiency will be largely determined by the time it takes to compute EAD estimates.

 Furthermore, the method of Zwaneveld and Verweij (2014a) is modelled in the modelling language GAMS and solved using the commercial solver CPLEX. While the method of Zwaneveld and Verweij (2014a) is, in principle, applicable to an arbitrary number of lines of defence, in practice the model code must be manually extended with new equations to implement any

additional lines of defence. While these extensions are trivial for anyone with experience in integer programming and GAMS, we believe that by automating these steps the threshold for using and applying these models can be lowered.

 Reducing the number of EAD estimates that will be computed can be done based on the principle of 'lazy evaluation', which delays calculations until they are actually required. However, 'lazy evaluation' requires a tight coupling between the EAD estimation and the economic optimisation routine. This tight coupling needs to be technically and organisationally pos-

sible. Organisationally, this tight coupling is possible in projects which are carried out by a single team combining all relevant disciplines; in the remainder of this research we assume that the organisational requirement is fulfilled. Technically, the economic optimisation routine needs to be able to dynamically call the EAD estimation function during its optimisation process. However, optimisation routines typically expect a pre-calculated set of data, which means an optimisation routine will need to

be modified in order to support 'lazy evaluation'. One such optimisation routine that can be easily implemented in a general programming language and adapted to use lazy evaluation is the shortest-path approach.

In the following section we further investigate the shortest-path approach in order to solve the problem of an economic optimisation for multiple interdependent flood defences. The aim of this paper is to develop a generic, computationally efficient approach for finding the economically optimal configuration of a flood defence system with an arbitrary number of interdependent flood defences which, for example, influence each other's EAD. The reliability and performance (in terms of number of EAD calculations) of finding economically optimal targets will be tested by comparing the results of the proposed method with a number of benchmark studies. We will accomplish this using the following approach:

- Computational efficiency will be primarily obtained by minimising the number of (time-consuming) Expected Annual Damage (EAD) computations in the algorithm until they are actually required (i.e. 'lazy evaluation')

- A generically applicable, flexible representation of the problem space will be presented which is able to use an arbitrary number of defences. Specifically, this entails generating a graph in an automated way based on an arbitrary number of interdependent flood defences

In section 2, a description of the application and a description of the applied algorithm are given. Implementation details, focused on the computational efficiency of the algorithm, are discussed in Section 3, as well as a list of potential future improvements to the algorithm. Next, the proposed approach is applied to some simplified case studies in Section 4, and is followed by a discussion (Section 5) regarding the relevance of the proposed approach. Finally, the results and experiences are concluded in Section 6.

## 2 An algorithm for flood defence systems with multiple interdependent flood defences

### 2.1 Programmatic representation of the solution space

A common choice to present optimisation problems is to use graph algorithms (Cormen, 2009). Regarding the economic optimisation of flood defences, this choice was also made in Zwaneveld and Verweij (2014b). An example of a graph for a single flood defence is shown in Figure 4. The graph shows the possible investments over time for a single flood defence. In this graph the vertices (dots) are the possible heights the flood defence can have at a certain point in time. In order to go the next point in time, edges are drawn which connect a vertex to all the possible vertices in the next point of time.

These points in time are not fixed; the amount and position can be altered to the needs of a particular problem. In practice, these points in time can be related to the (political) decision process of a particular problem: if the relevant flood defences are reviewed and (if necessary) reinforced every five years, it would make sense to have a graph that corresponds to these points in time.

Generally speaking, edges in a graph can be directed or undirected. However, steps backwards in time do not make sense for investment schemes. Therefore, only edges directed forward in time are used. The edge cost (or weight) of an edge is the total

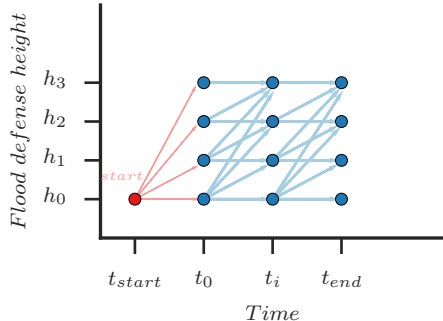

**Figure 4.** Graph where the vertices (dots) at each time step are connected via edges (arrows) to the next time step.

cost (EAD plus investment cost) of moving between the connected vertices. Furthermore, it is assumed that flood defences will not be intentionally decreased to a lower level, which is why, for example, there are no edges running from $h_1$ to $h_0$. The starting point of the graph is denoted with start in Figure 4 at time $t_{start}$ at a height equal to the current height ($h_0$).

In case of multiple flood defences, our method takes into account that flood defences can be interdependent and interact
with each other hydrodynamically. This means that the EAD of the system of defences can potentially be influenced by each defence, which also means that each combination of flood defence levels has to be considered relevant. For a graph with multiple interdependent flood defences, these combinations replace the height of a single flood defence on the y-axis in Figure 4. These combinations of heights for multiple flood defences can be obtained by computing the Cartesian product of the flood defence levels of all the involved defences. For $n$ flood defences, the Cartesian product equation for determining the
combinations is shown in Eq. 3:

$$\prod_{i=1}^{n} \boldsymbol{X}_i = \boldsymbol{X}_1 \times \ldots \times \boldsymbol{X}_n$$

$$= \{(x_1, \ldots, x_n) \,|\, x_1 \in \boldsymbol{X}_1, \ldots, x_n \in \boldsymbol{X}_n\} \quad (3)$$

where $\boldsymbol{X}_i$ is a vector containing all the flood defence levels of flood defence $i$, and $x_i$ is a realisation of vector $\boldsymbol{X}_i$ (i.e. a flood
defence level for flood defence $i$). If all vectors $\boldsymbol{X}_i$ are of the same length $y$, the total number of combinations will be $y^n$.

The number of relevant system combinations reduces significantly if each flood defence can be optimised independently of the other flood defences in the system. The assumption of independence can be made if none of the flood defences in a system have a (significant) influence on the EAD estimates of the other flood defences. The total number of system configurations under the independence assumption is $n \cdot y$, as each flood defence could then be optimised separately (e.g. using a graph per
flood defence similar to the graph shown in Figure 4. If only some defences are independent from other flood defences in the system, this is considered a special case of our approach. In that case, our method of using the Cartesian product is still valid and applicable, although it will result in larger than necessary graph. In case of some independent elements, a possible approach to reduce the size of the graph is discussed in Section 3.4.

Figure 5 shows an example of the Cartesian product for two flood defences where each flood defence has two possible heights. The graph in Figure 5 resembles the graph in Figure 4 for a single flood defence. Similar to Figure 4, edges in Figure 5 are only drawn to vertices containing sets of heights equal or greater than the set of heights in the vertex at the origin of the edge. However, because Figure 5 has two defences instead of one, the outgoing edges are slightly different when compared to Figure 4. For example, the height combination $h_{A0}, h_{B1}$ is never connected to $h_{A1}, h_{B0}$ (since that would correspond to a reduction in height for defence $B$).

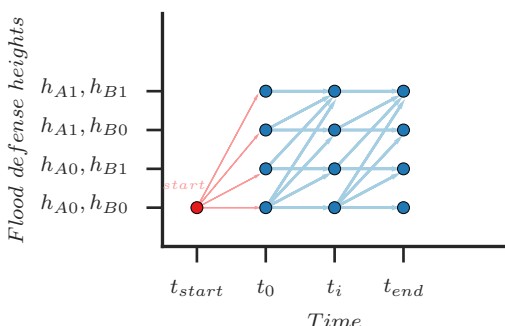

**Figure 5.** Graph with vertices (dots) and edges (arrows) for two defences ($A$ and $B$). Each defence has two possible heights.

## 2.2 Implementation of a graph algorithm

In general terms, a graph algorithm will iterate over vertices in a graph in an effort to find the path with the lowest costs between a given start and end vertex. However, in the graphs of Figure 4 and Figure 5 $t_{end}$ contains a number of possible end points, which means that the algorithm will need to find as many optimal paths as there are end points in the graph. In order to only have to run the algorithm once, a stop vertex is added; the graph of Figure 5 with an additional stop vertex is shown in Figure 6. The edges running towards this stop vertex are all given a weight of zero. Now, the algorithm only has to find a single optimal path between $t_{start}$ and $t_{stop}$. Why this is an efficient contribution is illustrated in Section 2.4.

The graph as shown in Figure 6 is a graph with directed non-negative edges. For this kind of graph, a number of algorithms can be used to find the shortest (optimal) path in a graph, for example: the Dijkstra algorithm (Dijkstra, 1959), the A* algorithm (Hart et al., 1968), and the Uniform Cost Search (e.g. (Verwer et al., 1989)). All three can be considered to be part of the family of best-first search algorithms, where both the Dijkstra and the Uniform Cost Search (UCS) algorithms can be seen as a special case of the A* algorithm.

Typically, the best-first search algorithms are implemented with a min-priority queue. A min-priority queue holds a sorted list of vertices, where the sorting is based on the cost of reaching that vertex from the start vertex; the vertex with the lowest cost is at the top of the queue. This list of vertices in the priority queue constitutes of, depending on the implementation, either all vertices in the graph (Dijkstra as implemented in Cormen (2009)), or only the vertices already visited by the graph algorithm (UCS). A comparison between the two algorithms can be found in Felner (2011), where the priority queue as implemented by

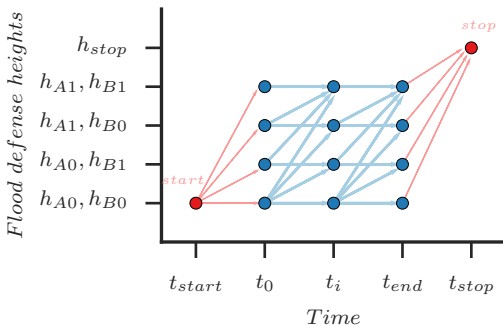

**Figure 6.** The graph of Figure 5 with an additional stop vertex.

UCS was found to be faster and using less memory. We consider this a relevant advantage, as the number of vertices can be large when using the Cartesian product of flood defence levels (Section 2.1). For this reason, we chose to implement the UCS algorithm.

In Eijgenraam et al. (2016), a dynamic programming approach was used, which is related to the shortest-path algorithms discussed thus far. However, the Dijkstra algorithm (and by extension the UCS and A* algorithms) are seen in Cormen (2009) as a part of the *greedy* shortest-path algorithms family, which in Cormen (2009) is clearly defined as a different type of algorithm than dynamic programming. Greedy algorithms are typically much faster than dynamic programming approaches, at the expense of not always finding the optimal solution (because less possible solutions are considered). The optimality condition is further discussed in Section 2.4. Nevertheless, because less possible solutions are considered in a greedy algorithm, this can also lead to a part of the graph never being visited by a greedy algorithm. Combined with 'lazy evaluation', this can lead to a significant reduction in the number of EAD calculations which are actually executed; see also Section 3.3.

Applying the UCS algorithm to a graph such as shown in Figure 6 begins with creating a priority queue which only contains the start vertex. After this initialization, the iteration process is started. Each iteration starts with taking out the vertex with the lowest cost known thus far from the priority queue (which is the top entry in the queue). Taking out means the optimal route (lowest cost) from the start vertex to this vertex now known. The vertex that has just been taken out of the priority queue is then queried in the graph to find all the connecting vertices in the next time step. Each connecting vertex is added to the priority queue if the vertex is not already in the queue. If the vertex already exists in the queue, the weight is only updated if the newly proposed cost is lower than the known cost so-far. Iteration continues until at the start of an iteration the stop vertex is the top entry in the priority queue. An actual example of the application of this algorithm will be elaborated in Section 2.3.

## 2.3 Example application of the algorithm in an economic optimisation

This section shows a simple example of an economic optimisation for a single flood defence. While this example uses a single flood defence for simplicity, the same principles apply for multiple flood defences. Regarding the investment costs and EAD estimates, if a vertex at $t_1$ is connected to another vertex with a larger height at $t_2$, it is assumed that the actual heightening

occurs at $t_1$. This leads to a slightly different graph than the conceptual implementation shown in Section 2.1 & 2.2, and is emphasized by drawing the edges of the figures in this example (i.e. Figures 7, 8 & 9) in a way which is visually more consistent with the timing of the investment decision.

The result of the first two iterations is shown in Figure 7, where the start vertex is labelled with the number 1. In this example, the start vertex is associated with a height of 4.25 meter and starts at $t = 0$, identical to vertex 2. Because the path to vertex 2 is the only possible path, vertex 2 is the only addition to the priority queue. In the next iteration, vertex 2 is taken out of the priority queue as it is the vertex with the lowest total cost. The total cost to reach vertex 2 is 0, because there was no heightening (height remains at 4.25 meter) and no time expired ($t_{start} = 0$); the EAD is zero because time needs to expire for risk to occur. From vertex 2, the number of possible next steps and associated total costs are computed and added to the priority queue, as illustrated in Figure 7. Note that the total costs to reach for example vertex 12 consists of the total cost from $t_{start}$ to $t = 100$, not just the cost from $t = 50$ to $t = 100$.

The algorithm will continue for a while, until the situation of Figure 8 is reached where vertex 24 is taken out of the priority queue. The new found total costs for vertex 29, 30 and 31 are not lower than the total cost for vertex 25, which means the algorithm takes a step back and continues from vertex 25. From vertex 25, vertex 30 and 31 are re-evaluated in Figure 9, where only vertex 30 results in lower costs than the existing options. This means that only vertex 30 is updated with the new, lower, total cost in the priority queue. Additionally, if hypothetically vertex 31 is the vertex with the lowest cost, the optimal path would revert back to using vertex 24 instead of vertex 25 (because the path from vertex 25 to 31 has higher costs than the path from vertex 24 to 31).

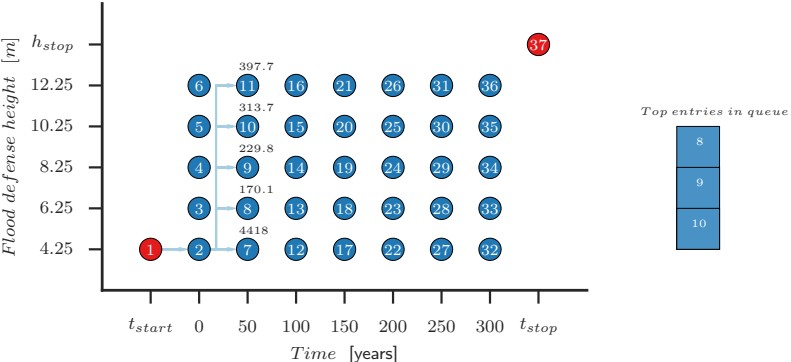

**Figure 7.** The first two iterations of the graph algorithm with a min-priority queue. The vertices available in the priority queue are those which have total costs above their respective vertices, and the first three entries are shown in the column on the right. Note that because the choice was made to connect the start vertex to vertex 2, vertices 3 - 6 will not be visited.

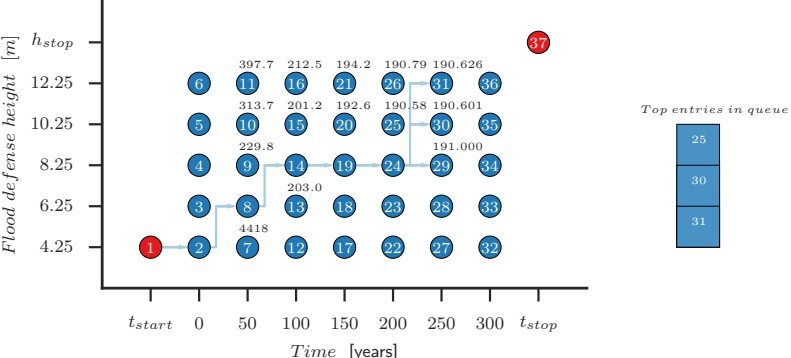

**Figure 8.** After six iterations with the graph algorithm, vertices 29, 30 and 31 are added to the priority queue. However, in this case the algorithm makes a step back in time, because vertex 25 is the item with the lowest total cost in the priority queue.

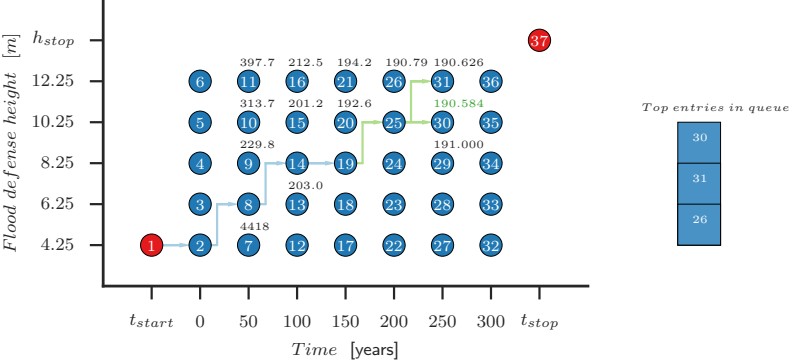

**Figure 9.** During iteration seven, the old path is abandoned, and an alternative path with vertex 25 instead of vertex 24 is taken. Vertices 30 and 31 are already in the priority queue (calculated from vertex 24), and will only get updated if the total costs from vertex 25 are lower (which is the case for vertex 30).

## 2.4 Global optimal solution

The UCS algorithm finds the shortest path in a graph, see for example Felner (2011) for a recent elaboration regarding the 'correctness' of the UCS algorithm or Gelperin (1977) for a proof regarding A* (UCS can be considered a special case of A*). What remains is whether the additional stop vertex of Section 2.2 leads to a potential heuristic solution or still to the optimal path. However, assuming that the optimal path towards the stop vertex is found, whichever vertex at $t_{end}$ is part of that optimal path has to be the optimal choice. Otherwise, the path towards the end vertex is not optimal, which contradicts the earlier mentioned proofs. In order to further test the performance of the proposed method, Section 4 will compare numeric results from our proposed method to other approaches. These approaches are known to give global optimal results.

## 2.5 Overview of the approach

A general overview of the approach discussed in the previous sections is shown in Figure 10. The method is composed of four steps: input, pre-processing, processing and post-processing. Of these steps, user interaction is only required at the input step. The rest of the steps run automatically. Specifically, the user needs to supply vectors of flood defence levels per flood defence, a time vector and a function which can calculate the cost of an edge in the graph. In the following steps, the graph is created (pre-process), the optimal path is found (process), and the optimal path is shown (post-process).

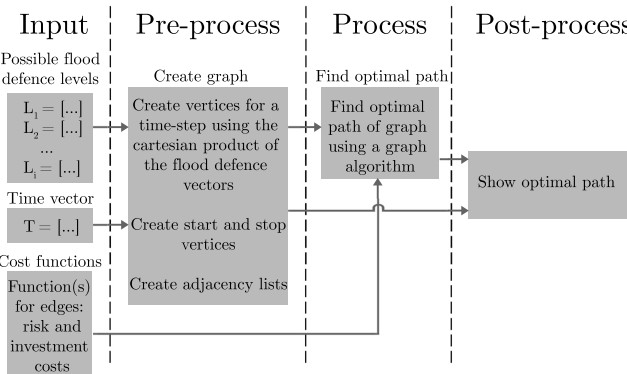

**Figure 10.** Overview of the approach using a graph and graph algorithm. In our approach, the graph algorithm is the UCS algorithm. The input column is the only part what the user should provide, the other steps run automatically.

## 3 Efficiency improvements

The economic optimisation of multiple interdependent flood defences, implemented in a graph using Section 2, can potentially lead to large numbers of vertices and even larger numbers of edges. For example, for eight interdependent flood defences with six possible heights the number of vertices per time step is approximately 1.68 million ($6^8$), while the number of edges per

time step is even larger at approximately 35 billion. For large problems such as these, storing all the possible vertices and edges would lead to huge data structures and a huge number of EAD calculations. This requires both an efficient implementation of the graph, and an efficient evaluation of EAD calculations (i.e. as few as possible). An efficient graph implementation is discussed in Sections 3.1 & 3.2, while the efficient evaluation of EAD calculations is discussed in Section 3.3. Potential further

efficiency improvements are discussed in Section 3.4.

## 3.1    Repetitiveness in lists of vertices

Even though the graphs of Section 2.1 can be classified as sparse graphs (number of edges is much smaller than the number of vertices squared, Cormen (2009)), the number of edges is still much larger than the number of vertices. Therefore, we first focused on data structures related to the edges of a graph. For sparse graphs, these are the adjacency lists: a group of vertices

connected via edges stemming from a source vertex in a previous time step. In these adjacency lists, repetitiveness can be found with respect to two aspects.

The first repetitive aspect is the similarity of adjacency lists for the same combination of flood defence levels at different time steps (except for the adjacency lists at $t_{end}$). In Figure 7, vertices with the same combination of flood defence levels at different time steps are for example vertices 2, 7 and 12. The adjacency lists for these vertices are shown in Figure 8, where it

is apparent that the adjacency list for the next time step can be found by adding an offset to the elements of the adjacency list of the current time step. For example, the adjacency list of vertex 2 can be turned into the adjacency list of vertex 7 by adding the total number of combinations in each time step (which is five in Figure 11)



**Figure 11.** The adjacency lists for vertices 7 and 12 of Figure 7 can be obtained by adding an offset to the adjacency list of vertex 2.

The second repetitive aspect is for adjacency lists between vertices in the same time step. Because the lowest vertex in each time step (e.g. vertices 2, 7, 12, 17, 22 and 27 in Figure 7) has outgoing edges running to each and every vertex in the next

time step, higher vertices (e.g. in Figure 7, vertex 8 is 'higher' than vertex 7) contain a subset of the adjacency list of the lowest vertex. In other words, outgoing edge lists in a single time step can be generated dynamically by shrinking the adjacency list of the lowest vertex in a time step. This is shown in Figure 12.

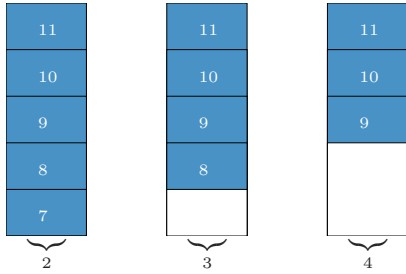

**Figure 12.** The adjacency lists for vertices 3 and 4 of Figure 7 are reduced sets of the adjacency list for vertex 2.

The combination of these two repetitive characteristics results in that only a single adjacency list needs to be stored in memory (i.e. the adjacency list of the lowest vertex in the first time step). This single adjacency list can be adapted to most vertices in the graph by means of offsetting and shrinking the stored adjacency list. Notable exceptions are the adjacency lists for the vertices at $t_{end}$, but the adjacency lists for these vertices are already known and only contain the stop vertex.

## 3.2 Conditionally removing edge connections

Besides reducing the size of the data structures associated with a graph, the adjacency list associated with a vertex can also be reduced under certain conditions. Typically, the time between improvements in flood defences is large (in the order of 50 years), due to either high (fixed and variable) costs associated with investments in flood defences, or long planning periods (Zwaneveld and Verweij, 2014b). Therefore, if one or multiple flood defences have been strengthened recently, the adjacency list can be reduced to only contain vertices that keep the recently strengthened flood defence(s) at the current level(s). However, this so called 'waiting time' before new investments are considered has to be chosen with care, because the waiting time should not influence the optimal time between investments. Nevertheless, a correctly chosen waiting time can greatly improve the run time of the algorithm, because of the significant reduction in number of edges that need to be evaluated. This reduction is shown in Figure 13, where the total number of visited vertices is plotted as a function of the 'waiting time'; the underlying problem that is solved by the algorithm is the same problem as shown in Section 4.3.

## 3.3 Reducing the number of EAD calculations

In the overview of Figure 10 it is implied that the EAD calculations belonging to an edge are only carried out when that edge is visited by the graph algorithm. Provided that a graph algorithm does not visit all vertices, delaying EAD calculations belonging to an edge until that edge is visited leads to less EAD calculations than the total number of possible EAD calculations in a particular graph. In contrast, if EAD (or more generally, cost) calculations are done before a graph algorithm is initialised, all possible EAD calculations need to be calculated beforehand.

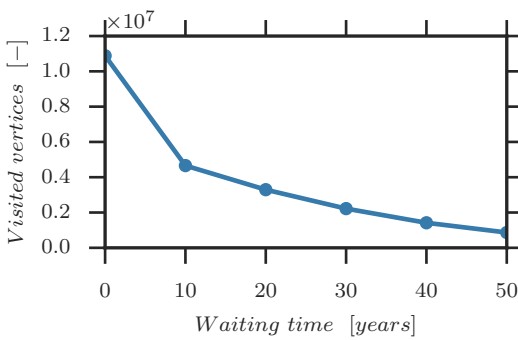

**Figure 13.** Total number of visited vertices as a function of the waiting time for the example of Section 4.3.

As an example, Figure 14 shows the number of times each vertex is visited is in the example of Section 2.2. The majority of the vertices in Figure 14 get visited once, but a significant proportion is never visited by the graph algorithm; these vertices have a zero above their indices. A small proportion of the vertices, specifically vertices 30 and 31, are visited twice; the reason for this re-visiting can be seen in Figure 9. To avoid completely re-doing cost calculations upon a revisit, parts of a calculation can be cached in order to reduce the computational penalty incurred by revisiting vertices. The total number of possible EAD calculations is the number of years multiplied with the number of options on the y-axis; for Figure 14 this leads to a total number of EAD calculations of 1505 (or $301 \cdot 5$). Because a number of vertices do not get visited by the algorithm, the number of actual executed EAD calculations goes down to 1000, or approximately 66% of all possible EAD calculations.

Furthermore, the number of EAD calculations can be further reduced by using the 'waiting time' of Section 3.2. In Section 3.2, it was found that a minimum waiting time between investments will lead to less edges being evaluated by the algorithm. This also implies that less EAD calculations will be executed. Using the same example as in Figure 13, the reduction in the percentage of actual executed EAD calculations is given as a function of the waiting time in Figure 15. Between using a waiting of 0 years (i.e. no minimum waiting time at all) and a waiting time of 50 years the number of EAD calculations goes down from approximately 60% to 40% for the example of Section 4.3.

## 3.4 Potential improvements and special cases

Further improvements can be made both to the graph implementation and to the implementation of the algorithm. The algorithm was implemented as a single process; a performance improvement might be found by utilizing parallel programming. The first place where parallel programming could be beneficial is the loop over an adjacency list. This is because the potentially expensive EAD calculations are done as part of determining an edge weight. Therefore, parallelising the loop over an adjacency list over multiple computational nodes can lead to significant performance improvements.

Furthermore, regarding the graph implementation, a special case is a flood defence system which has independent flood defences. Section 2.1 uses the Cartesian product of flood defence options, which has the underlying notion that all flood defences are interdependent. If some flood defences are independent (i.e. the defences protect different, independent areas),

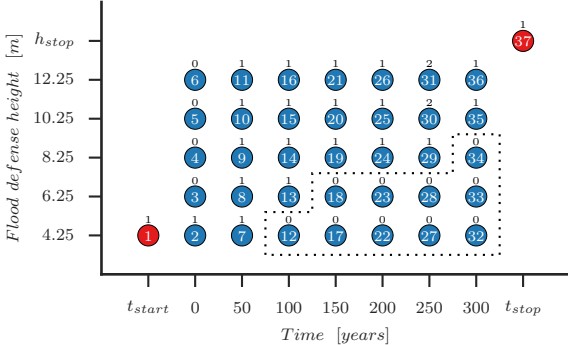

**Figure 14.** Number of times each vertex is visited by the algorithm for the example in Section 2.2. The dotted area emphasizes that a part of the graph is never visited, while vertex 30 and 31 get visited twice.

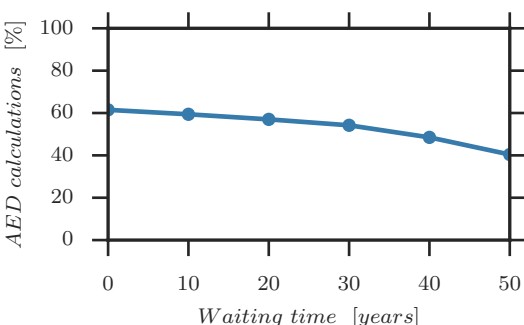

**Figure 15.** Percentage of actual executed EAD calculations as a function of the waiting time for the example of Section 4.3.

this leads to an inefficient graph. The independency of flood defences can be used in an adapted graph representation in order to get an efficient graph. While we did not implement this, a way to solve this inefficiency for the system in Figure 16 is shown conceptually in Figure 17, which uses 'subgraphs' to reduce the number of combinations.

These subgraphs are small graphs which only contain the number of strengthening options for a single defence for a single time period (e.g. in Figure 17, from $t_{i-1}$ to $t_i$). Additionally, the subgraphs take into account what the level is of the influential defences (e.g. in Figure 17, the front defence $B$ is the only influential defence for the rear defences). The use of subgraphs leads to a smaller number of combinations, as the Cartesian product would have resulted in a total number of 5120 ($5 \cdot 4^5$) vertices per time step. With subgraphs, the number of vertices per time step is reduced to 100 ($5 \cdot 5 \cdot 4$).

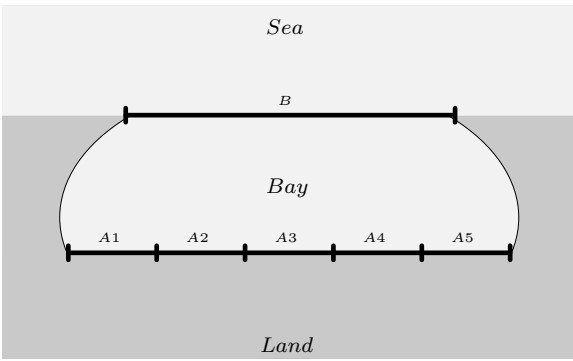

**Figure 16.** A top view of a system with a front line defence ($B$, five possible safety levels) and five rear defences ($A1 - A5$, each has four possible safety levels). The front defence influences the rear defences, but the rear defences do not influence each other.

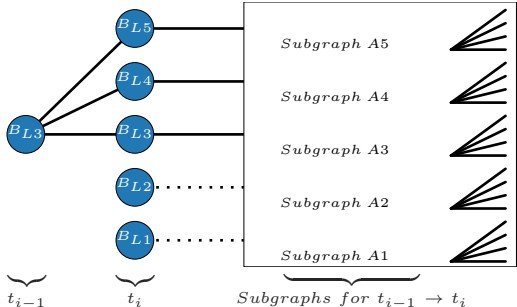

**Figure 17.** Part of the graph belonging to the system of Figure 16 for the period $t_{i-1}$ to $t_i$. Because the rear defences do not influence each other, subgraphs are used for the rear defences.

## 4    Results for simplified flood defence systems

In order to test the performance of the proposed algorithm versus some existing approaches, three cases are investigated. For simplicity, these three cases will be based upon a common set of investment and EAD relations, as well as a common set of input values. The values and symbols used in this section are largely copied from (Eijgenraam, 2006, page 34) and reproduced

5    in Table 1, with only minimal changes. These EAD and investment cost relations consist of simple formulations which were specifically chosen for exhibiting the approach, for ease of reproducibility, and for showing the efficiency regarding the number of EAD calculations. In practice, EAD estimates can be quite complex and/or have a high computational burden, especially when flood defences are modelled to have hydrodynamic interactions with each other. For example, a single EAD estimate for a complex flood defence system with interdependencies can take hours (Klerk et al., 2014) or even days (Courage et al., 2013).

10    The common set of investment ($I$) and EAD (or flood risk cost, $R$) relations are similar to the relations used with the data of Table 1 in Eijgenraam (2006). The sum of the investment cost and EAD is the total cost, which needs to be minimised in order

**Table 1.** Variables and values taken from Eijgenraam (2006) for the EAD and investment equations in this section. NLG refers to the currency used in the Netherlands prior to the euro.

| Name | Unit | Symbol | Value |
|---|---|---|---|
| Height above mean sea level, base | cm | $H_0$ | 425 |
| Annual exceedance probability belonging to $H_0$ | - | $P_0$ | 0.0038 |
| Parameter exponential distribution water level | 1/cm | $\alpha$ | 0.026 |
| Increase water level | cm/year | $\eta$ | 1 |
| Damage by flooding in 1953 | $10^6$ NLG | $V_0$ | 20000 |
| Economic growth | 1/year | $\gamma$ | 0.02 |
| Rate of interest (real) | 1/year | $\delta$ | 0.04 |
| Variable costs of investment | $10^6$ NLG/cm | $C_v$ | 0.42 |
| Fixed costs of investment | $10^6$ NLG | $C_f$ | 61.7 |
| Heightening of the flood defence at time $t$ | cm | $u_t$ | - |
| Height of the flood defence at time $t$ | cm | $H_t$ | - |

to get economically optimal safety targets:

$$\text{Total Cost} = \int_0^\infty R(t)dt + \sum_{t=0}^\infty I(t) \tag{4}$$

$$R(t) = P_0 e^{-\alpha(H_t - H_0 - \eta t)} V_0 e^{\gamma t} e^{-\delta t} \tag{5}$$

$$I(t) = (C_v u_t + C_f \text{sign}(u_t)) e^{-\delta t} \tag{6}$$

5    where sign($u_t$) is used to prevent fixed costs in case there is no heightening $u_t$. This sign($u_t$) function returns zero if the heightening $u_t$ is equal to zero, and returns one when the heightening $u_t$ is larger than zero.

## 4.1   Single flood defence

For a single flood defence, with the values of Table 1, an analytical solution can be found in (Eijgenraam, 2006, page 35). This solution consists out of an initial dike height increase coupled with a periodical, constant dike increase over an infinite

time horizon. The numerical results were re-calculated with the solution listed in Eijgenraam et al. (2016), and resulted in an immediate initial increase of 235 centimetres with a periodical increase of 129 centimetres every 73 years.

Because the approach introduced in this paper is a numerical approach, a finite time period had to be used instead of an infinite time horizon. Similar to Zwaneveld and Verweij (2014b), we choose to use a time period of 300 years with, for this
application, steps of one year. The possible heights were discretized using a range starting from 425 to 1225 cm, with steps of one centimetre. Note that these step sizes (and dimensions) were deliberately chosen to be on par with the accuracy level of the analytical solution. In practice, these step sizes would probably be too detailed for the practical attainable accuracy in flood defence construction (see also Zwaneveld and Verweij (2014b)). Furthermore, the total number of possible EAD calculations in this problem is 241,101 (or $801 \cdot 301$). Of these, 137,971 were actually executed by the UCS algorithm, which corresponds
to using only 57% of all possible EAD calculations. Increasing the 'waiting time' to 50 years did not affect the solution but did reduce the percentage of executed EAD calculations down to 43%.

A comparison of the results found using the algorithm and the analytical solution is shown in Figure 18. The algorithm found an initial increase of 235 cm, with three additional increases in height at 73 years apart. These three were found to be 129 cm, 130 cm, and 132 cm. The last increase is different from the analytical solution, and can be attributed to being close to the end
of the time horizon. A finite time horizon implies that there is no EAD beyond the time horizon, which explains why there is no investment found by the algorithm in year 292. To compensate for the lack of an investment in year 292, the investment in year 219 is slightly larger. This explanation is supported by results with a time horizon of 400 years, where the heightening in year 219 changes to an expected 129 centimetres. These deviations near to the time horizon underline that if a certain point in time is considered relevant, the used time horizon should stretch significantly beyond that point in time. However, this is a
general problem with all numerical methods, because of the required finite time horizon, and not a specific issue related to the approach proposed in this paper. Furthermore, in practice this problem can be circumvented by setting the time horizon used in the algorithm to sufficiently exceed the practically required time horizon.

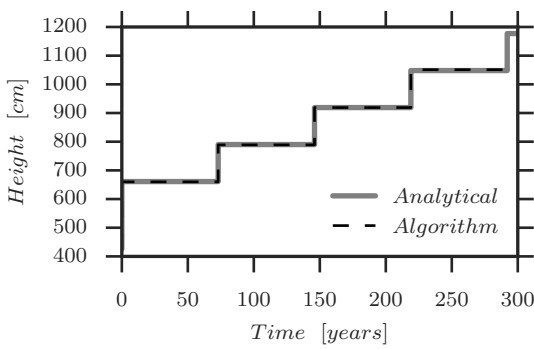

**Figure 18.** The investment scheme found using the algorithm is almost identical to the analytical solution.

## 4.2 Two independent flood defences

In the next example two defences are investigated using the graph algorithm, both with the same characteristics as the single flood defence in the previous section. However, the step size for the heights is increased to 20 cm in order to test the response of the algorithm to larger step sizes. Expected is that, despite the less detailed step size, the investment scheme for both defences should be identical to each other and close to the analytical solution provided in the previous section.

Indeed, the results of the algorithm, illustrated in Figure 19, show that both defences are initially increased with 240 cm, while in both year 75 and 143 the defences are increased with 120 cm, and finally in year 212 with 140 cm. Clearly, the larger step size in height leads to larger differences when compared to the analytical solution. Nevertheless, any overshoot/undershoot of the height is 'repaired' in the duration between investments, keeping the solution of the optimal path stable and close to the analytical solution. Furthermore, the total number of possible EAD calculations in this problem is 24,682 (or $2 \cdot 41 \cdot 301$). Of these, 14,510 were actually executed by the UCS algorithm, which corresponds to using only 59% of all possible EAD calculations. If a 'waiting time' of 50 years is used, the solution is unaffected but the percentage of executed EAD calculations goes down to 48%.

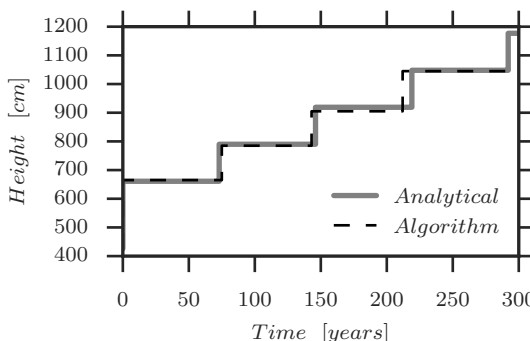

**Figure 19.** The two (independent) flood defences have an identical solution with the approach proposed in this paper, and are (even with the usage of larger step sizes) good approximations of the known analytical solution.

## 4.3 Two dependent flood defences

The final case is similar to the case with two independent flood defences, however the second defence is now dependent on the performance of the first defence. This dependency is illustrated in Figure 16, and is a simplified version of the case discussed in Dupuits et al. (2016).

The dependency between the defences in Figure 20 is implemented by adapting the EAD equation of Eq. 5 as follows:

$$R\left(t\right) = \left(P_1 P_{2|1} + \left(1 - P_1\right) P_{2|\bar{1}}\right) V_0 e^{\gamma t} e^{-\delta t} \tag{7}$$

$$P_i = P_0 e^{-\alpha_i \left(H_{i,t} - H_0 - \eta t\right)} \tag{8}$$

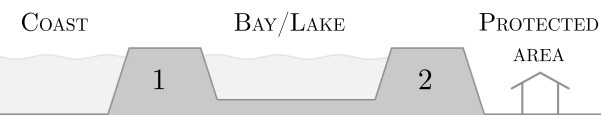

**Figure 20.** A coastal system with two lines of defence. This figure is an adaption from an illustration found in Dupuits et al. (2016).

where $P_i$ is a generic formulation used for the failure probabilities $P_1$, $P_{2|1}$ and $P_{2|\bar{1}}$. The probabilities and are the failure probabilities of the second defence, dependent on the failure ($P_{2|1}$) or non-failure ($P_{2|\bar{1}}$) of the first defence, where the failure probability of the first defence is denoted by $P_1$. Similarly, the investment equation in Eq. 6 is expanded to include different costs for the two lines of defence:

$$I(t) = (C_{v1}u_1 + C_f \text{sign}(u_1))e^{-\delta t}$$

$$+ (C_{v2}u_2 + C_f \text{sign}(u_2))e^{-\delta t} \quad (9)$$

The new variables used in Eqs. 7, 8 & 9 are listed in Table 2. The solution found with the approach proposed in this paper was checked with the method proposed in Zwaneveld and Verweij (2014a); the outcomes of both methods were found to be identical and are shown in Figure 21. Furthermore, the total number of possible EAD calculations in this problem is 505,981 (or $41^2 \cdot 301$). Of these, 311,190 were actually executed by the UCS algorithm, which corresponds to using only 62% of all possible EAD calculations. If a 'waiting time' of 50 years is used, the solution is unaffected but the percentage of executed EAD calculations goes down to 40%.

**Table 2.** Additional variables used in Eqs. 7, 8 & 9, complementary to Table 1.

| Name | Unit | Symbol | Value |
|------|------|--------|-------|
| Annual exceedance probability belonging to $H_o$ | - | $P_0$ | 0.01 |
| Exponential parameter for defence 1 | 1/cm | $\alpha_1$ | 0.026 |
| Exponential parameter for defence 2 for $P_{2|\bar{1}}$ | 1/cm | $\alpha_{1|\bar{2}}$ | 0.052 |
| Exponential parameter for defence 2 for $P_{2|1}$ | 1/cm | $\alpha_{1|2}$ | 0.026 |
| Variable costs of investment for defence 1 | $10^6$ NLG/cm | $C_{v1}$ | 0.21 |
| Variable costs of investment for defence 2 | $10^6$ NLG/cm | $C_{v2}$ | 0.42 |

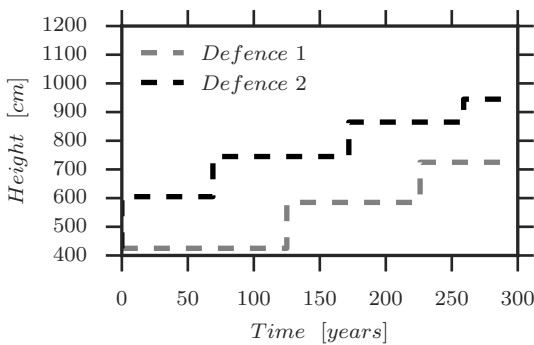

**Figure 21.** Optimal investment schemes for the case with two interdependent flood defences.

## 5 Discussion

The proposed approach (see also Figure 10) in this paper is based on a best-first graph algorithm, which is relatively easy to implement in most general or scientific programming languages. In our opinion, this is a significant advantage over linear programming algorithms, especially for those who are not familiar with the implementations of linear programming as pro-
posed by Zwaneveld and Verweij (2014a). Although the application area is the same as for Zwaneveld and Verweij (2014a), notable differences are present between the two approaches. The approach of Zwaneveld and Verweij (2014a) is capable of including both interdependent & independent flood defences and focused on finding the proven economically optimal solution quickly given pre-calculated EAD estimates and investment costs. Our approach focuses on flood defence systems with mostly interdependent flood defences (though Section 3.4 does discuss a possible efficient extension to mostly independent flood de-
fences) and computational costly EAD calculations. Therefore, the focus of our approach is on reducing the number of actually executed EAD calculations (compared to pre-calculating all possible EAD estimates).

An inherent problem of working with flood defence systems where most, if not all, elements are dependent on each other, is that the number of system combinations grows exponentially with the number of interdependent flood defences. The sheer number of combinations means that the total number of interdependent flood defences should probably be kept below ten. This
is nothing more than a rule of thumb based on our experience running the best-first graph algorithm on a consumer laptop. The true maximum depends on a number of factors: the number of height options per defence, the performance of the particular implementation of the proposed approach, the computational cost of the associated EAD functions and the computational power of the computer used.

Even though all examples in this research make use of flood defence heights, this was only done to illustrate the approach.
Other measures besides flood defences can be incorporated as well in the graph that is used to find the optimal solution. While this is not a unique feature of our approach (i.e. any graph-based approach can do this), it is a relevant point for the viability of practical applications. For example, if a retention area is considered (as illustrated in Figure 22), a list with possible sizes of the retention area could also be used in the approach of Figure 10: in principle, as long as a measure has a number of options

or levels in increasing order that can be quantified and monetised, it can be included in the approach. This makes the actual application range much wider than flood defence systems with only height-dependent flood defences such as levees or (storm surge) barriers.

The proposed approach works best if the type of each flood defence is known and singular. In the case that a number of different defence types are considered for the same flood defence, it would be better to do an optimisation run per type of defence. An example of this would be the choice between a closure dam or a storm surge barrier at the same location. In this case, the algorithm should be run twice, first with a closure dam and then with a storm surge barrier. This should result in two optimal configurations (one with a closure dam, the other with a storm surge barrier), which can then be compared using the same metric, for example their benefit-cost ratios.

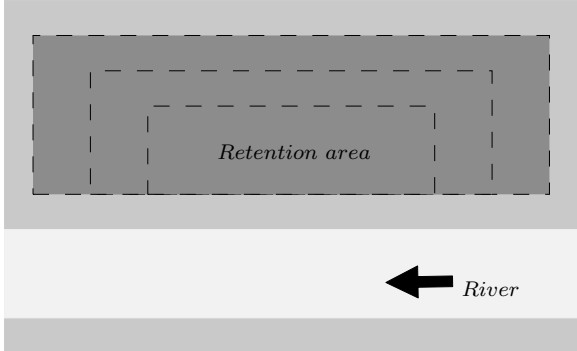

**Figure 22.** A retention area can also be optimised using the approach proposed in this paper. In this example, the surface area of the retention basin is used instead of the height of a flood defence.

## 6 Conclusions

This paper presented a generic, computationally efficient approach for finding the economically optimal configuration of a flood defence system with an arbitrary number interdependent flood defences. Computational efficiency was achieved by delaying EAD (Expected Annual Damage) calculations until they are actually needed in an optimisation routine (i.e. 'lazy evaluation'), which leads to a reduction in the number of EAD calculations that need to be done. In the examples shown in this paper, the reduction in number of EAD calculations was at least 40%. This is a significant and relevant reduction, as the EAD calculations relevant for this approach often have a high computational cost. This is especially the case when multiple flood defences interact with each other hydrodynamically in a larger flood defence system.

The approach presented in this paper uses a best-first graph algorithm, which is simple to implement and advances existing shortest-path implementations for economic optimisation of interdependent flood defence systems. Furthermore, the approach is flexible towards the number and type of flood defences because the graph representation shown in this paper can trivially accommodate an arbitrary number of interdependent flood defences. The proposed approach utilizes the repetitive properties

of the graphs in order to efficiently store the representation of the graph in memory. In case independent flood defences are present in a system, the proposed approach of generating a graph can be adapted to a more efficient method which makes use of the attractive properties of independence. To that end, a concept has been proposed which reduces the size of the graphs.

Assuming that the graph and combinations of flood defences are portrayed correctly, the best-first graph algorithm has been proven in literature to return the shortest (or optimal) path in a graph. To corroborate this for our implementation and intended application, the method was tested on a number of benchmark problems with known solutions. The tests show that indeed the optimal path is found with the approach proposed in this paper, which justifies the conclusion that the implementation was done correctly.

*Author contributions.* TEXT

*Competing interests.* TEXT

*Disclaimer.* TEXT

*Acknowledgements.* We are grateful for the financial support of the Dutch Technology Foundation STW, which is part of the Netherlands Organization for Scientific Research and is partly funded by the Ministry of Economic Affairs. Furthermore, we are grateful to Peter Zwaneveld and Gerard Verweij of the CPB for sharing their ILP model with us. We are also would like to acknowledge two anonymous reviewers for their comments which improved this manuscript.

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
