# Peer review of "Economically optimal safety targets for interdependent flood defences in a graph-based approach with an efficient evaluation of expected annual damage estimates"

_Natural Hazards and Earth System Sciences, 2017_

## Referee Comment (RC1) · Anonymous Referee #1 · 5 Mar 2017

The paper provides a coherent narrative and is clearly within the scope of NHESS. It also provides a scientific background to how engineers can systematically explore a multi-dimensional space for optimal solutions using a method unknown to many. As such it is publishable.

My review is based on my background knowledge which is more related to the traditional cost-benefit analyses in relation to risk-based design. So please bear with me if there are things I have misunderstood. On the other hand I have done exactly the same as the authors using traditional economic tools in relation to risk-based design.

I think the paper should be rewritten to improve clarity. Therefore I only have overall comments.
The paper, and in particular the Introduction section, is not very well written for the reader not already familiar with the thinking of the authors. Assumptions about prior knowledge on Dutch design criteria are very high, previous work is not introduced as more than a reference (sometimes even to studies in Dutch).

The authors seem to use the term risk to characterize probabilities and economic loss interchangeably. Please define and use a clear notation. This could be done in relation to Equation 1 which in poorly defined. The problem is encapsulated in the sentence on page 2, line 17.

The authors rightly state (e.g. page 3 line 5) that the major work in relation to risk-based design is calculation of the residual risk (in monetary terms) by a complex procedure involving complex hydrological and hydraulic calculations and subsequent calculation of loss of vulnerable assets. However, I cannot see how calculation of the edges as outlined on e.g. page 4 line 5 can be done without such calculations. Indeed this is also stated on page 7, line 5. So I see a reduction in the required number of calculations in comparison to LP, but at least as computationally demanding than traditional Cost-Benefit Analyses. It is not difficult to set up the mathematical framework for optimization within economics that can identify economically optimal solutions if the risk can be formulated in a simple equation such as the authors do in their examples (e.g. Eq 3).

I would prefer if the extension involving several dikes heights to be optimized simultaneously were introduced using multiple dimensions. Since the paper only discusses two dimensions it should be straight forward also to show graphically. It will make comparison to marginal economic studies on efficiency of alternative measures quite apparent. Still, the visualization and structured approach to identify optimal trajectories makes the approach valuable. NHESSD

---

## Referee Comment (RC2) · Anonymous Referee #2 · 10 Mar 2017

In a supplement , I provide an pdf-document of my referee report.

Unforetunately, I cannot copy the plain text of my referee report in this 'box'

Please also note the supplement to this comment:
http://www.nat-hazards-earth-syst-sci-discuss.net/nhess-2017-12/nhess-2017-12-RC2-supplement.pdf

---

## Author Comment (AC1)

**Answers by the authors on the referee report of reviewer #1**

Date: 08/05/2017

We took the original text of the review and divided it up into smaller pieces, which we answered. The original text by reviewer #1 is presented in blue, italic text, while our answers are numbered and in black.

*The paper provides a coherent narrative and is clearly within the scope of NHESS. It also provides a scientific background to how engineers can systematically explore a multi-dimensional space for optimal solutions using a method unknown to many. As such it is publishable.*

*My review is based on my background knowledge which is more related to the traditional cost-benefit analyses in relation to risk-based design. So please bear with me if there are things I have misunderstood. On the other hand I have done exactly the same as the authors using traditional economic tools in relation to risk-based design.*
*I think the paper should be rewritten to improve clarity. Therefore I only have overall comments.*

*The paper, and in particular the Introduction section, is not very well written for the reader not already familiar with the thinking of the authors. Assumptions about prior knowledge on Dutch design criteria are very high, previous work is not introduced as more than a reference (sometimes even to studies in Dutch).*

1. Thank you for this comment. We will expand the introduction to include a better introduction of relevant Dutch methods and design criteria.

*The authors seem to use the term risk to characterize probabilities and economic loss interchangeably. Please define and use a clear notation. This could be done in relation to Equation 1 which in poorly defined. The problem is encapsulated in the sentence on page 2, line 17.*

2. Thank you for this comment. We make the use of risk/probabilities/economic loss consistent and clear.

*The authors rightly state (e.g. page 3 line 5) that the major work in relation to risk-based design is calculation of the residual risk (in monetary terms) by a complex procedure involving complex hydrological and hydraulic calculations and subsequent calculation of loss of vulnerable assets. However, I cannot see how calculation of the edges as outlined on e.g. page 4 line 5 can be done without such calculations. Indeed this is also stated on page 7, line 5. So I see a reduction in the required number of calculations in comparison to LP, but at least as computationally demanding than traditional Cost- Benefit Analyses. It is not difficult to set up the mathematical framework for optimization within economics that can identify economically optimal solutions if the risk can be formulated in a simple equation such as the authors do in their examples (e.g. Eq 3).*

3. Thank you for this comment. We tried to answer it in three parts:
   - Edge calculations: It is correct that each edge is associated with a potential risk calculation. The reduction in number of calculations is indeed compared to I(L)P.
   - Traditional C/B analyses: we assume that by traditional C/B analyses, the reviewer refers to marginal C/B analyses (i.e. those that optimise flood defences separately and independently of hydrodynamic interactions in a larger system of flood defences). We will further clarify that our proposed method of looking at the whole flood defence system makes sense if hydrodynamic interactions are expected to lead to significantly different flood risk estimates. If the flood risk estimates are approximately the same with and without hydrodynamic

interactions, the economic optimisation might be done just as well (and possibly more efficiently) by looking at each flood defence independently.

- Simple equations: We purposefully chose these simple equations in order to focus on the approach and not on the examples. However, these simple equations should not be seen as representative of the risk calculations we have in mind. In follow-up research, we have a more complex case study in which hydrodynamic interactions are explicitly modelled. However, if we were to include this (or a similar) case study in this paper it would need to have either a lengthy description (muddying the focus of the paper) or a reference to future (unpublished) work which would make the case hard to reproduce. Nevertheless, we can (and will) add a description that these equations are simplified for the purpose of this paper, and can be replaced by (for example) hydrodynamic simulations in a Monte Carlo setting.

We will make these points clearer in the paper.

*I would prefer if the extension involving several dikes heights to be optimized simultaneously were introduced using multiple dimensions. Since the paper only discusses two dimensions it should be straight forward also to show graphically. It will make comparison to marginal economic studies on efficiency of alternative measures quite apparent. Still, the visualization and structured approach to identify optimal trajectories makes the approach valuable.*

4. Thank you for this comment. We interpreted this comment as that we need to explicitly show and describe in that the approach is applicable to more than two lines of defence. We will alter the examples section accordingly.

---

## Author Response (AR1)

**Response by the authors to the comments of reviewer 1**

Date: 20/07/2017

We took the original text of the review and divided it up into smaller pieces, which we answered. The original text by reviewer 1 is presented in blue, italic text, while our responses are numbered and in black.

Any references to lines in the manuscript are made to the manuscript which has the visible changes compared to the old manuscript. The manuscript with the visible changes can be found at the end of this document.

The paper provides a coherent narrative and is clearly within the scope of NHESS. It also provides a scientific background to how engineers can systematically explore a multidimensional space for optimal solutions using a method unknown to many. As such it is publishable.

My review is based on my background knowledge which is more related to the traditional cost-benefit analyses in relation to risk-based design. So please bear with me if there are things I have misunderstood. On the other hand I have done exactly the same as the authors using traditional economic tools in relation to risk-based design. I think the paper should be rewritten to improve clarity. Therefore Lonly have overall

*I think the paper should be rewritten to improve clarity. Therefore I only have overall comments.*

The paper, and in particular the Introduction section, is not very well written for the reader not already familiar with the thinking of the authors. Assumptions about prior knowledge on Dutch design criteria are very high, previous work is not introduced as more than a reference (sometimes even to studies in Dutch).

- 1. Thank you for this comment. We have expanded the introduction to include a better introduction of the economic optimisation of flood defences as it used in the Netherlands. Specifically:
  - The addition of Figures 1 and 2, the expansion of Eq.1 and the addition of Eq.2
  - The additions/changes on Page 1, lines 15-23 and Page 2, lines 1-15
  - The study in Dutch is now accompanied with a reference to a chapter from a PhD thesis, which is in English (page 4, line 8)
  - A more complete overview of relevant literature (e.g. page 3 lines 17-22, page 4 lines 1-8)

The authors seem to use the term risk to characterize probabilities and economic loss interchangeably. Please define and use a clear notation. This could be done in relation to Equation 1 which in poorly defined. The problem is encapsulated in the sentence on page 2, line 17.

- 2. Thank you for this comment. We have clarified our definition of risk/probabilities/economic loss.
  - Specifically, see Eq.1 and Eq.2 and the description of these equations on page 1 lines 15-23.
  - The introduction of using AED, or Annual Expected Damage as an alternative for the annual expected risk cost (page 1, lines 19-23, now used throughout the paper).

The authors rightly state (e.g. page 3 line 5) that the major work in relation to risk-based design is calculation of the residual risk (in monetary terms) by a complex procedure involving complex hydrological and hydraulic calculations and subsequent calculation of loss of vulnerable assets. However, I cannot see how calculation of the edges as outlined on e.g. page 4 line 5 can be done without such calculations. Indeed this is also stated on page 7, line 5. So I see a reduction in the required number of calculations in comparison to LP, but at least as computationally demanding than traditional Cost- Benefit Analyses. It is not difficult to set up the mathematical framework for optimization within economics that can identify

**economically optimal solutions if the risk can be formulated in a simple equation such as the authors do in their examples (e.g. Eq 3).**

- 3. Thank you for this comment. We tried to answer it in three parts:
  - Edge calculations: It is correct that each edge is associated with potential risk calculations. The main challenge is there to reduce the required number of edges "visited", to save valuable computation time. The reduction in number of calculations is indeed compared to I(L)P. We have tried to make this clear throughout the paper, particularly in:
    - Page 4, lines 9-15
    - The introduction of section 3 (page 12, lines 4-12)
    - Section 3.3 on page 14-15 and Figures 14, 15
    - $\circ~$  By adding the percentage of actual executed calculations to the examples in Sections 4.1-4.3
    - In the first paragraph of the Discussion on Page 21-22.
    - By focusing on the reduction of cost calculations in the first paragraph of the Conclusions on page 21
  - Traditional C/B analyses: we assume that by traditional C/B analyses, the reviewer refers to marginal C/B analyses (i.e. those that optimise flood defences separately and independently of hydrodynamic interactions in a larger system of flood defences). We have clarified that our proposed method of looking at the whole flood defence system makes sense if hydrodynamic interactions are expected to lead to significantly different flood risk estimates. If the flood risk estimates are approximately the same with and without hydrodynamic interactions, the economic optimisation might be done just as well (and possibly more efficiently) by looking at each flood defence independently. See also:
    - Equation 3 on page 6, and the description on page 6, lines 13-19 and page 7 lines 1-12.
  - Simple equations: We purposefully chose these simple equations in order to focus on the approach and not on the examples. However, these simple equations should not be seen as representative of the computational costly risk calculations we have in mind. In follow-up research, we have a more complex case study in which hydrodynamic interactions are explicitly modelled. However, if we were to include that (or a similar) case study in this paper it would need to have either a lengthy description (muddying the focus of the paper) or a reference to future (unpublished) work which would make the case hard to reproduce. Nevertheless, we have added a description that these equations are simplified for the purpose of this paper, and can be replaced by (for example) more complex hydrodynamic models and probabilistic computation methods.
    - Specifically, see Page 17, lines 2-6.

I would prefer if the extension involving several dikes heights to be optimized simultaneously were introduced using multiple dimensions. Since the paper only discusses two dimensions it should be straight forward also to show graphically. It will make comparison to marginal economic studies on efficiency of alternative measures quite apparent. Still, the visualization and structured approach to identify optimal trajectories makes the approach valuable.

- 4. Thank you for this comment. We interpreted this comment as that we need to explicitly show and describe in that the approach is applicable to more than two lines of defence.
  - We have done so by adding Equation 3 (page 6) and the description on Page 6 (lines 13-19) and page 7 (lines 1-12).

**Response by the authors to the comments of reviewer 2**

**Date: 20/07/2017**

We took the original text of the review and divided it up into smaller pieces, which we answered. The original text by reviewer 2 is presented in blue, italic text, while our responses are numbered and in black. The headings ("General comment" and "Specific comments") are retained from the original review text.

Any references to lines in the manuscript are made to the manuscript which has the visible changes compared to the old manuscript. The manuscript with the visible changes can be found at the end of this document.

**General comment:**

Before reading this referee comment, the reader must be aware of the fact that the authors of this paper actively asked me to referee their paper. I thank them for this opportunity and making me aware of this and previous papers. I have had a meeting with two authors of this paper to discuss my first impressions. This referee comment benefitted from the insight the authors provided me in this meeting.

In this general comment I will state that this paper copies earlier work. No proper references are made to this work. Hence, this paper doesn't meet minimal scientific standards. I will provide references to plenty published reports and articles to support my claim. In the Netherlands, many involved experts, including many full professors at various Dutch universities (which wrote and published referee reports on this earlier work or supported the development of earlier work) can be asked to confirm my claim.

The main claim by the authors that a shortest path/dynamic programming approach (previously presented and discusses by other authors) to solve economic optimal dike heightening is 'advantageous' needs to be elaborated a lot more. Many previously stated and published arguments against dynamic programming are not mentioned. Moreover, the scientific ambition of this paper is not clear. Furthermore, no calculations are presented by Dupuits et al. (2017) to support their claim. I will provide arguments for this claim.

We thank the reviewer for taking time to sit down with us.

**We do not comment on these first three paragraphs, as it seems that all the mentioned issues return in a more detailed form in the remainder of the review document.**

This paper copies the approach by Zwaneveld & Verweij (2014a) for finding an optimal configuration for interdependent lines of flood defences. In Zwaneveld (2012; section 1.1; in 2014 provided to the authors by email) several approaches are discusses to solve this model. This modelling approach by Zwaneveld & Verweij (2014a, 2014b) including graph-based (shortest path or minimum cost flow problems) solution approaches and the preferred ILP approach was earlier copied and described by Yuceoglu (2015, Chapter 5: Safe Dike heights in the Netherlands). This PhD-thesis builds on the work by Zwaneveld and Verweij (2014a, 2014b) and discusses graph –based algorithms to solve the so-called Diqe-Opt model (see later for a discussion of this model). Zwaneveld & Verweij (2014a) identify several algorithms to solve the problem both to proven optimality and to solve the problem heuristically (with the advantage that computing times remain limited). Zwaneveld & verweij92014a0 aplly their model to ral world problem instances to support crucial Cabinet decisions for the Netherlands.

- We share a similar approach as Zwaneveld & Verweij (2014a), by using graphs to model the problem. However, contrary to the approach of Zwaneveld & Verweij (2014a), we use a greedy algorithm to solve the shortest path problem instead of modelling it as an I(L)P model. The reason for using a greedy algorithm is, among other reasons, an attempt to reduce the number of risk calculations (see also answer 3 for a more detailed answer). Therefore, we disagree with the use of the word 'copy'. Nevertheless, we did miss that in the appendix of Zwaneveld & Verweij (2014a) the problem was already identified as a graph problem. We therefore improved the paper:
  - The entire introduction (pages 1-5) has been rewritten with help of suggestions from reviewer 2, and should now contain a more complete overview of related publications
  - Acknowledgements now also express gratitude to Zwaneveld & Verweij for sharing their model

I apply and explain an algorithm to solve the problem to proven optimality. Their algorithms requires hardyly any programmng efforts and little solution time ('less than one minute or so'). The ideas to solve the problem heuristically were not implemented in practice due to the fact that the algorithm to solve the problem to proven optimality was superior according to Zwaneveld and Verweij (2014a).

- 2) Based on our own experience, finding the shortest path in a graph with a greedy algorithm can be explained intuitively for most engineers working in the field of flood risk. This explanation is one of the motivations of writing Section 2. It is the opinion of the authors that an IP model (specifically the model as proposed by Zwaneveld and Verweij (2014a)) requires at least a basic knowledge of integer programming models. Even if the model code of the IP model is available to a user, the model code will need to be expanded if that user wants to add an additional line of defence. Granted, the extension is relatively straightforward (provided the user understands the linear programming model), but it is not a "blind copy-paste action". Contrary to this, our approach builds (and solves) the graph automatically for an arbitrary number of lines of defence and, given the model code, only needs inputs. This has been further explained in the paper:
  - The introduction, where this issue is now introduced (page 4, lines 9-24 & page 5 lines 1-5)
  - Equation 3 on page 6, which is the basis for automatically building the graph, and Sections 3.1 and 3.2 which describe how the entire graph can be represented with only a fraction of the entire graph. Note that we expanded the description of Section 3 on page 12 with a clearer separation between the graph in-memory representation (3.1 and 3.2) and the reduction in risk cost calculations. (3.3)
- 3) Furthermore, we clearly state in our aim that we want to achieve computational efficiency by means of reducing the number of risk calculations. In the IP approach by Zwaneveld and Verweij (2014a), the flood risk calculations are not considered as a part of the solving time. Instead, the risk calculations are assumed to have be carried out beforehand in their approach. If all flood risk calculations (and other calculations) are done beforehand, and the only issue at hand is the solving time of the algorithm, the IP approach will be more efficient than the greedy algorithm. We don't contest this in the paper, because we do not consider the solving time of the graph algorithm as dominant. We think there are plenty of scenarios where the flood risk calculation time is dominant (and even limiting). In that case, we have shown that a greedy algorithm (i.e. one that does not necessarily visits all vertices) coupled with an efficient evaluation of the risk estimates (i.e. only calculate the risk if a vertex is actually visited, as opposed to calculating the risk for all existing vertices), is expected to result in fewer risk calculations for most situations. How many calculations will be saved depends wholly on the case study characteristics and discretization. This has been further explained in the paper:
  - We rewrote the approach in the introduction (Page 5, lines 13-22) to make sure that our definition of computational efficiency is a reduced number of risk cost calculations, not the efficiency of the algorithm itself. Similarly, the conclusions have been updated (page 23-24).
  - A similar addition as mentioned in the previous point has been made to the abstract

• Risk cost calculations are assumed to have a (much) higher computational burden than the optimisation algorithm, which is now stated multiple times in the paper (e.g. see page 4 lines 9-15).

In line with Brekelmans et al (2012), Eijgenraam et al, (2010; in revised version published as: Eijgenraam et al. 2016) a dynamic programming (read: shortest path approach) is identified in Zwaneveld (2012) as one of the options to solve the model. Zwaneveld and Verweij (2014a, Annex A, Figure A; 2014b) and Zwaneveld (2012) contribute to these earlier papers by identifying that the dike optimization problem can be seen as a graph based problem. For example, Zwaneveld and Verweij (2014b, p.29) state that the the dike optimization model 'satisfies the most fundamental of all network flow problems (Ahuja et al., 1993), namely the minimum cost flow model'. This point was missed by earlier published work and also by Dupuits et al. 2017. Dupuits et al. (2017) present a graph based representation which is identical to the graph representation of Zwaneveld and Verweij (2014b). For example, compare Figure A in Zwaneveld and Verweij (2014b, p. 29) with the figures of Dupuits et al. 2017. They are clearly (almost ) identical.

- 4) Regarding the figures containing graphs being almost identical: we think that graphs with this kind of structure always look almost identical. Nevertheless, as already mentioned in 1), we added references to Zwaneveld & Verweij regarding seeing the flood defences optimisation problem as a graph. Regarding the dynamic programming approach in the listed papers (Eijgenraam et al, (2010), Eijgenraam et al, (2016)): dynamic programming is mentioned there without specifying which dynamic programming algorithm is actually used. Dynamic programming can actually entail a number of algorithms (Cormen (2009)). Furthermore, in Cormen (2009), a clear distinction is made between dynamic programming and greedy algorithms. We explicitly mention that we use a greedy algorithm. Therefore, (part) of our contribution is to use a greedy algorithm instead of a dynamic programming approach. This is further elaborated upon in the paper:
  - See page 5, line 31 for an additional reference in Section 2.
  - See page 8, line 17-24 for the discussion regarding dynamic programming.

Unfortunately, this paper by Dupuits et al. (2017) does not clearly refer to these previous papers and reports which they copy and build upon. In my opinion, this paper needs a thorough revision to correctly and clearly refer to the work of previous mentioned authors to meet minimal scientific standards.

**5) see 1), where we address these citation issues.**

This paper states in the introduction that "However, existing cost-benefit analyses tend to focus on flood defences with independent lines of (Kind 2014), or are not readily generically applicable (e.g. Zwaneveld and Verweij (2014a). Therefore, the aim of this paper is to find general, computationally efficient approach... with arbitrary number of lines"

The authors do not mention the fact that Zwaneveld and Verweij (2014a, including background papers), Bos and Zwaneveld (2012) and Zwaneveld and Verweij (2016, UK CPB discussion paper on previous Dutch reports) do present for the first time a generic, computationally approach to assess dependent flood defense systems whit arbitrary number of flood defense lines.

6) We choose the words "readily generically applicable" with the argumentation of 2) in mind. In 2), we acknowledge that the IP model is extensible, but extending it requires at least some editing. Furthermore, see 1) for citation issues.

Moreover, these authors do apply their approach in a real world environment and under time pressure to obtain economic optimal flood protection policy measure for the Lake IJssel region (including many dependent dike rings and barrier dams). Dupuits et al. (2017) are aware of this approach and solution method since they apply it in section 4.3. Zwaneveld and Verweij kindly provided Dupuits et al. (2017) with their programming code and data to allow scientific reuse of their earlier work.

Although this approach is not yet published in in a UK written scientific journal (the authors are working on it, see Zwaneveld and Verweij, 2016)), the scientific quality had been assessed by two different committees with professors and other experts (see Donders et al., 2013; Van Ierland et al., 2014). This was due to the fact that very important hydrological and economic policy decisions are based upon the application of the Diqe-Opt model (in Bos and Zwaneveld 2012; Zwaneveld and Verweij 2014a). The Ministry of Infrastructure and Environment had to be sure about the quality of the Diqe-Opt model and the two reports. Documents are published on the UK and Dutch based CPB-website. A few documents are also presented to the Dutch Parliament: they can also be found at the website of the Dutch Parliament.

The latter 2014-committee of four professors at Dutch universities conclude in Dutch: "Het CPB heeft met deze studie een belangrijke stap vooruit gezet in het onderzoek naar waterveiligheid. Het is een indrukwekkende studie waarin zeer veel hydrologische en economische kennis op een prachtige manier wordt samengebracht. Met name het meenemen van afhankelijkheden in de overstromingskansen van dijken is een belangrijke innovatie. Het ontwikkelde model Diqe-Opt is een vernieuwend en zeer nuttig instrument" [UK translation: "CPB has made with this study an important step forward in the search for water safety. It is an impressive study in which very many hydrological and economic knowledge is combined in a wonderful way. In particular, the inclusion of dependencies in the flood dikes of opportunities is an important innovation. The developed model Diqe-Opt is an innovative and very useful tool"". Note that Zwaneveld and Verweij (2014) name their 3 generally applicable method: Diqe-Opt. Due to the generally applicable of the Diqe-Opt approach the model is by request from the Ministry of Infrastructure and the Environment being transferred to hydrological consultancy company Deltares. Deltares can use the model as long as proper references are made to earlier CPB-work by Zwaneveld and Verweij. The Dutch institutions setting of the CPB (employer of Zwaneveld and Verweij) prohibit these activities by CPB. This innovative Diqe-Opt approach was also recognized by a recent Dutch handbook on water safety (ENW, 2016, Literature list to Chapter 4).

Hence, Dupuits et al. (2017) should state that they copy the Diqe-Opt model by Zwaneveld and Verweij (2014a, 2014b) instead that the "aim to find an ... approach". Proper and clear references are missing towards this earlier work in the starting sections of this paper.

7) We are thankful for sharing the model with us. We already referred to using it in Section 4.3, and we have extended our gratitude to the acknowledgements. Regarding the suggested change for the aim, missing references and the usage of 'copy': see 1) where we address the same comment.

*Dupuits et al (2017) present a shortest path algorithm to the problem definition as presented by Zwaneveld and Verweij (2014a).*

 8) We interpreted this as a similar comment (regarding the problem definition/approach) as one that is already answered by our answer in 1). Therefore, see 1) for our answer.

For the cases presented in section 4.1 and 4,2, almost identical and probably more efficient dynamic programming approaches (shortest path approaches) are presented in Eijgenraam et al. (2010) and Brekelmans et al. (2012). Proper references should be made to this earlier work.

9) According Cormen et al (2009), greedy algorithms typically need less computational time than dynamic programming approaches. From that perspective, we do not see how dynamic programming can be more efficient than greedy algorithms. We have clarified this in the paper; see also 4).

Furthermore, we do not think we need to show an extensive comparison with existing solutions, as we do not present our approach as a competitor to these existing methods. See also our answer in 16) where we answer the same comment.

I do not see the value added by the shortest path approach of section 2 and section3 in addition to these two papers.

- 10) Because we believe we use a different algorithm to solve the shortest path approach, see also 4) where we mention the difference between dynamic programming and greedy algorithms.
- 11) The purpose of section 2 is to show the basic functioning of the greedy algorithm with respect to a (very) simplified problem, in combination with a graph. We believe this helps to create a fundamental understanding for engineers who are not familiar with the concept; as the intended target audience might not be as familiar with graph algorithms as an operations research audience. Furthermore, understanding how a greedy algorithm 'moves' through a graph (along with lazy evaluation) is essential for achieving the reduction in risk cost calculations.
  - Specifically, see page 8 lines 22-24
- 12) Section 2.4 contains references which are relevant for the used greedy algorithm, and provides context to whether or not the shortest path (optimal solution) of a graph with non-negative edge weights is found, which is important for any optimisation algorithm. Furthermore, Section 3.3 explains (based on section 2) how the risk cost calculations are reduced.
  - See also the additions on page 15, lines 2-11, Figure 14 and Figure 15.

Zwaneveld and Verweij (2014b, Annex A) present an alternative approach for these two cases in section 4.1 and 4,2 based upon an ILP-model. Applying this approach requires no programming effort whatsoever since user friendly software can be used to model the problem.

**13) We addressed the implementation of I(L)P models in 2).**

No efforts are required to solve the stated model since standard LP/IP solvers can be used. Note that an LPsolver is at present a plug in tool in Microsoft Excel and modelling languages as CPLEX, GAMS and AIMMS are easily available. Free and easy to use solvers are easily available.

- 14) CPLEX, GAMS and AIMMS are all tools that are geared towards applications with, for example, I(L)P models such as used in Zwaneveld and Verweij (2014). This requires at least basic knowledge of this kind of model, and knowledge of the specific tools. See also 2). This is now also mentioned in the paper on page 4, line 16-20.
- 15) GAMS and AIMMS are algebraic modelling systems with proprietary licenses. CPLEX is a solver with a proprietary license. Free solvers, at least at the time of writing, don't scale well with thousands of decision variables and will have a hard time to solve the problems as discussed in Zwaneveld and Verweij (2014). On the contrary, our approach can be used in any general-purpose programming language, such as for example the freely available open-source languages Python and Julia. Inherently, this means that threshold for application and use is lower than with proprietary licenses.

We did not think it was necessary to include the above in the paper, as our primary

goal is an efficient computation time (by reducing the amount of risk cost calculations), and not presenting a direct competitor for the Zwaneveld and Verweij (2014) model; see also the changes mentioned in 3).

The authors should mention in section 4.1 and 4.2 the use of these competitive and in some cases almost identical approach and should compare it with their approach. This comparison was already presented in Zwaneveld and Verweij (2014b, Annex A). They provide arguments and conclude that an ILP-approach is by far more preferable than a dynamic programming approach. Dupuits et al. (2017) should – as a minimum- discuss this work by Zwaneveld and Verweij (2014b).

- 16) The purpose of the (simplified) examples in section 4 is to show that the greedy algorithm works in terms of reducing risk cost calculations; not to show a comparison between various shortest path algorithms or I(L)P models. See also 3) for the context of our method. We measure computational efficiency in the number of risk cost calculations actually evaluated, versus the total number of possible risk cost calculations in a graph. This has been further elaborated upon in the paper:
  - In the abstract and in the introduction, specifically in the approach (page 5)
  - Section 3.3, particularly page 15 lines 2-11
  - In the examples of section 4.1-4.3 (page 18 lines 6-9, page 19 lines 17-20, page 21 lines 7-10)

Furthermore, it is unclear why Dupuits et al 2017 conclude that an dynamical programming approach is 'relatively easy'. From a discussion with the authors, I learned that their intention and scientific ambition is to present a heuristic approach to solve the model by Zwaneveld & Verweij (2014a). Although heuristic approaches are presented in Zwaneveld & Verweij (2014a, 2016), these were not yet implemented. The advantages of this heuristic approach is to reduce calculation time with the disadvantage that a non-optimal solution is found. Furthermore, to help the authors, some persons may prefer a dynamic programming approach over an ILP approach. I also learned that a dynamic programming approach is easier to understand for many people than an ILP-approach. Therefore, Zwaneveld & Verweij(2014a, 2014b) always present their model as a graph problem and then introduce that they prefer to solve this graph problem to optimality by using an ILP-approach.

- 17) Our aim is to use a greedy algorithm in order to reduce the number of risk cost estimates actually executed (see also 4 for the difference between dynamic programming and greedy algorithms). We think that the UCS algorithm, as discussed in for example Fellner (2011), is easy to implement because the fundamental concepts related to UCS area taught in basic algorithm classes (Fellner, 2011). Furthermore, we consider the actual implementation in code as easy because of the brevity of the algorithm (see Fellner (2011), about 10 lines of pseudo code), coupled with the fact that the target audience (civil engineers) is more likely to be familiar with general purpose programming languages (see also 15)) than IP models (see also 14)). The optimal path is found for a graph using a greedy algorithm, if implemented as explained in Section 2.
  - Figure 10 already described explicitly that the user only needs to provide the inputs, provided that the greedy algorithm and automated graph generation have been implemented and shared with the user.

However, the ambition by the authors, as I learned from personal communication with them, doesn't meet their statement on page 5 of Dupuits et al. (2017) : "For our applications, we did not come up with a heuristic

function, which reduces the choice of a graph alfgorithm to either Dijkstra or UCS". Hence, this requires more explanation by the authors. I cannot see why both claims are valid.

We interpreted this comment as that the reviewer sees a greedy algorithm as a heuristic. This interpretation is answered in 18). However, we meant to discuss greedy algorithms with an optional heuristic function, which is clarified in 19).

- 18) An attribute of greedy algorithms is that, while efficient, they sometimes don't find the shortest path in a problem. We assume this is what the reviewer means with a heuristic. However, in Section 2.4 (see also 12)) we think we provide some references containing arguments for the greedy algorithm that it does find the optimal path for the discussed application area. Therefore, we refrained from calling the greedy algorithm a heuristic.
- 19) A heuristic function, as mentioned in the paper, relates to a subset of greedy algorithms. Heuristic functions can be used with a greedy algorithm in order to give additional information in an attempt to speed up the finding of the shortest path. A graph algorithm which can use such a heuristic is the A\* algorithm. If the heuristic function always returns zero (i.e. what we called no heuristic function), the A\* algorithm reduces to the Dijkstra algorithm or the closely related UCS algorithm:
  - Ultimately, we thought this was a non-essential detail and removed it to prevent any further confusion (page 8, lines 7-8).

For the case presented in section 4.3 proper references should be made that this is a simplified version of Zwaneveld and Verweij (2014a).

20) We do not see how a case as simple and general as shown in section 4.3, which in this general form can be found in many coastal areas around the world, should be referred to as a simplified version of the case presented in Zwaneveld and Verweij (2014a). We think that the fact that we use (and refer to) the method of Zwaneveld and Verweij (2014a) as a benchmark for the correct answer implies that the method of Zwaneveld and Verweij (2014a) can be applied to the case study as well. In addition, see our reply in 1).

Again, an explicit discussion of the pro's and cons of solving the model by Zwaneveld and Verweij (2014b) by using an ILP-approach and their approach should be presented. Zwaneveld and Verweij (2014a) and Zwaneveld (2012) do present such a comparison and they conclude that the ILP-solution approach is superior to dynamic programming (or: shortest path) for real-life applications. This earlier assessment should be presented. Why do Dupuits et al. (2017) conclude the opposite?

21) After careful re-reading of our own words, we cannot find any evidence that suggests we even compare the performance of ILP model to our greedy algorithm, let alone conclude that the greedy algorithm is superior. The only qualifications we make is that the outcomes are equal, with the Zwaneveld and Verweij (2014a) method being the benchmark, and that less risk calculations are done. See also 3) and 16).

Why do the authors present only 'toy problems instances' which can easily be solved using existing approaches?

- 22) Because a more complex case study would shift focus from the topic, which is to explain the application of graphs in combination with a greedy algorithm in the context of computationally expensive risk calculations (A more complex case study would require a significant amount of introduction regarding the assumed hydrodynamic interaction between multiple lines of defence, for example). The main point of the simplified examples is to show, in conjunction with earlier mentioned points in 12) and 18) and section 2.4, that the optimal path is found and that risk cost calculations are saved. A more complex case study will be an integral part of a follow-up research.
  - We have improved this description in the paper on page 17, lines 2-6.

**Zwaneveld and Verweij (2014a) and Bos and Zwaneveld (2012) were capable of solving very large real-time problem instances given very short research leadtime and research capacity.**

23) Our approach has different goals than the mentioned approaches. This was already touched upon in 2), 3), 14) and 15). Primarily: the computational cost of risk calculations is an issue, secondary: knowledge of IP models (and specifically the model by Zwaneveld and Verweij) is not commonly present among the target audience. For our intended use (see also our re-phrased aim of the paper) and the intended target audience, we consider the replies of 2), 3), 14) and 15) highly relevant.

Moreover, setting up a dynamic programming algorithm is requires very substantial programming efforts as is clear from section 2 and 3 from this paper. The approach by Zwaneveld and Verweij (2014a) requires only the code "SOLVE DIQE-OPT MODEL USING CPLEX" to obtain the proven optimal solution. Hence, the claim that a dynamic programming is 'more easy' than a ILP-approach by Zwaneveld and Verweij (2014a) is not valid or – at best - not properly motivated in my opinion.

24) We do not agree with this statement. See also 23) for a summary about the relevance of our approach. From our own experience, re-creating the model of Zwaneveld and Verweij (2014a) took a roughly equal amount of programming (We had to recreate the model based on the model description/formulas due to an absence of the necessary proprietary licenses for GAMS and CPLEX). This excludes the amount of time it took to familiarize ourselves with the IP model by Zwaneveld and Verweij and its inner workings. Furthermore, even though our approach did require a programming effort, we can share this code with anyone who has access to a computer, because it was coded using a freely available open-source language. Therefore, in principle, no additional coding effort is required by third parties regarding implementation of graphs and the greedy algorithm. See also 17).

**Specific comments**

**Section 2.1:**

The representation of the problems copies the approach by Zwaneveld and Verweij (2014a) and Zwaneveld and Verweij (2014b). Especially the graphs in this section are strikingly identical to Figure A from Zwaneveld and Verweij (2014a) See also almost identical figures in Yuceoglu (2015). Also references should also be made to Dynamic programming approach by Eijgenraam et al. (2010) and Brekelman et al. (2012) which seems to be mathematically identical. Proper references are missing to this earlier work. Dupuits et al. (2017) should clearly state that they copy previous work.

The cases presented in paragraph 4.1 (single flood defense) and 4.2 (independent lines of defences) can be solved by the dynamic programming (or shortest path approach) which is extensively discussed in Eijgenraam et al (2010) (a revised version of this paper was published as Eijgenraam et al 2016) and briefly discussed in Brekelmans et al. (2012). Zwaneveld & Verweij (2014b, 'paper under revise and resubmit') make the point in Annex A that these shortest path problems can be much easier solved to proven optimality using LP-relaxation or IP-model formulation. All this should be mentioned.

25) This remark seems to repeat a number of earlier made remarks, which we answered in our earlier replies. Addendum: Our focus is not to find the most efficient graph algorithm, our focus is to find a graph algorithm which handles risk calculations efficiently. In our opinion, this makes the requested addition of earlier made comparisons between dynamic programming and IP models non-relevant (at least not relevant for this paper). See also 3).

**Section 2 and 3:**

The presented approach is basically the well known shortest path algorithm. The discussion should can be deleted or removed to an electronic companion . I do not see any scientific added value in comparison with earlier work by Brekelmans et al (2012), Eijgenraam et al (2010) and the large literature of shortest path problems and dynamic programming. I personally prefer to refer to the well-written UK –based Wikipedia discussion of the subject(see Zwaneveld, 2012).

26) As mentioned in previous points:

- In our opinion we don't use dynamic programming (we use a greedy algorithm).
- Furthermore, we think section 2 serves as an essential introduction for nonexperts in the domain of graph optimization, which we think is relevant given the journal's audience. Furthermore, it serves as a foundation for explaining how the number of risk cost calculations can be reduced. See also 10, 11, 12.

The claim that repetitiveness of vertices is in most cases incorrect. Note that vertex 12 represent a later year than vertex 7 (see Figure 9). Due to yearly increases of economic growth and flood probabilities al risk calculation has to be calculated again. Hence, vertex 7 and 12 are not identical and no calculation time is saved.

- 27) We disagree with this comment. Risk calculations are not mentioned in section 3.1.We use the repetitive characteristics to reduce the size of the data structures belonging to the vertices and edges. This has been improved in the paper:
  - Section 3, particularly on page 12 lines 9-12.

Note that Brekelmans et al. (2012, p.1343) state that a simple 'homogenous case can be conveniently solved using dynamic programming. Unfortunately, this is not possible for the nonhomogenous cases, because the state space explodes.... We show how the nonhomogeneous diek height problem can be solved as a MINLPproblem.' A more or less similar statement by Brekelmans et al. (2012, p. 1345): "Unfortunately, the state space grows too large ....which implies that the dynamic programming approach is not applicable". This is the – very good- reason why Brekelmans et al. (2012) prefer their MINLP approach.

Dupuits et al (2017) do not properly discuss this exploding problem, i.e. exploding state spaces and exponential calculation time of all sorts in the problem size. Nor do they refer to these previously mentioned authors which did identify this problem before.

28) We do not use dynamic programming, we use a greedy algorithm. Furthermore, we believe the state space explosion can be partly negated with the help of 3.2 and with section 3.4 (if 3.4 is applicable), because these techniques can reduce the amount of edges and vertices (and therefore the computational time). Furthermore, state space explosion is inevitable if all flood defences are assumed to depend on each other (assumption repeated throughout the paper, see for example page 6, lines 14-16.)

From the paper I have got the impression that they apply a shortest path algorithm to solve the problem to proven optimality given – theoretically- computing time which are exponential in the problem size. From personal communication with the authors, I did get a different impression, namely that they aim to present a non-optimal solution approach given limited computing time. The authors should clarify that ambition.

A more or less similar remark holds for the algorithm. From the paper I get the impression that they implemented the algorithm themselves to find a proven optimal solution. From personal communication, I did get the impression that they use standard plug-in heuristic procedures to solve the graph. Hence, no programming effort whatsoever is required. The latter would make their approach of course more easy to use but also make their algorithm less innovative. The authors should clarify their ambition.

29) Unfortunately, it seems like we didn't make this sufficiently clear during our personal communication. We use an existing greedy shortest path algorithm, which we tried to implement in an efficient manner for the particular problem of economic optimization of multiple lines of defence. Efficiency was sought primarily by reducing the number of risk cost calculations (i.e. Section 3.3). See many of our earlier replies, for example 12), 18), 19) and 22), for further clarifications and replies.

**Section 5:**

The authors state that Kind (2014) proposes an linear programming approach. This is incorrect. Kind (2014) doesn't propose any method. He uses the approach by Brekelmans et al. (2012), which is an MINLP-approach. The IP-approach was proposed by Zwaneveld and Verweij (2014b, 'paper in revise and resubmit' to an academic journal). An IP-approach is not identical to a linear programming approach.

**30) Thank you for these suggestions. We have improved the introduction, see also 1).**

The claim that the application area is roughly similar to Zwaneveld and Verweij (2014a) is incorrect. The application area is completely identical and copied from Zwaneveld and Verweij (2014a). Furthermore, reference should be made that dynamic programming/shortest approaches of cases in section 4.1 and 4.2 to Eijgenraam et al (2010) and Brekelmans et al. (2012). And to heuristic ideas (and some attempts) to solve the dike height problem in previous work by Eijgenraam, Brekelmans and Den Hertog and Zwaneveld & Verweij (2014a, 2014b, 2016)

31) We refer to our answer of 1) regarding the suggested use of the word 'copy'. See also our answer in 3) for our aim, which (although similar) emphasizes different aspects than Zwaneveld and Verweij. Therefore, we refrained from using words such as 'identical' and 'copy'. We believe our proposed approach is complementary to the approach by Zwaneveld and Verweij (and other earlier proposed methods); it is not meant as a replacement.

*Lines* 17-22 *Page* 17: *The authors should mentioned that fact that the approach by Zwaneveld and Verweij* (2014b) was especially develop to include other flood defence systems than height-dependent dikes.

**32) Thank you for this suggestion. See page 22, line 17-19 for the relevant addition to the discussion.**

The fact that Dupuits et al (2017) can also include these approaches is a direct consequence of the fact that they copy the approach by Zwaneveld and Verweij (2014a, 2014b) and, therefore, both have identical application areas. The Diqe-Opt model was already used to asses many of these alternative flood defence systems in Bos and Zwaneveld (2012) and Zwaneveld and Verweij (2014a). See also Donders et al. (2013) and van Ierland et al. (2014)

33) Regarding the use of qualifications like 'copy' and 'identical application areas': See 1) and 3). See also 32) for properly referring to using alternative flood defence systems with Zwaneveld and Verweij (2014).

**Section 6**

The authors claim that it is an advantage that 'their approach do not need pre-calculate risk which linear programming approaches do'. However, the IP-approach by Zwaneveld and Verweij(2014a) – again this is NOT a linear programming approach – indeed does require risk estimates in a pre-processing step. In addition, stating that risk calculation can be performed 'on the fly' is complete impractical in a real-world setting of Zwaneveld & Verweij (2014a), Bos and Zwaneveld (2012), Brekelmans et al (2012) and Eijgenraam et al. (2016), since it requires in general running hydrological models. Hence, the approach by Dupuits et al. (2017) requires in each iteration to consult a hydrological experts to run their model and to report the result back. Doing these calculations in a pre-processing step as advocated by Zwaneveld and Verweij (2014a and 2014b) and Bos and Zwaneveld (2012) has very significant practical advantages. For real-world instances, risk calculation were no problem whatsoever in the approach by Zwaneveld & Verweij (2014a, 2014b), Brekelmans et al. (2012) and Eijgenraam et al. (2016).This argumentation is missing in this section.

- 34) We disagree that each iteration requires consulting a hydrological expert. In followup research, we are calculating risks 'on the fly' in a case study. We do not know the details of the referred to risk calculations, but we can predict that in the setting of this paper (multiple lines of defence of which the risk of a downstream defence is assumed to be dependent on all upstream defences), risk calculations will not be computationally cheap. This has been emphasized in the paper:
  - Page 4, lines 9-15
  - Page 4 lines 21-23 and page 5 lines 1-5

*Finally, the claim by Depuits et al. (2017) that their approach requires less risk calculation than the graph-based ILP—approach by Zwaneveld and Verweij(2014a) is not supported by calculations.*

- 35) See 3) where we argue why we cannot predict the amount of savings (depends on the case). We did mention the savings of the Section 2 example, specifically in figure 12 of section 3.3. This has been further expanded:
  - An expanded description of the number of saved risk cost calculations on page 15, lines 2-11
  - The number of risk calculations have been added to the examples of Section 4 (see also 16).

**Using graphs to find economically optimal safety targets for multiple lines of flood defences**

Egidius Johanna Cassianus Dupuits1, Ferdinand Lennaert Machiel Diermanse2, and Matthijs Kok1 1Delft University of Technology, Faculty of Civil Engineering and Geosciences, P.O. Box 5048, 2600 GA Delft, Netherlands 2Deltares Unit Inland Water Systems, department of Flood Risk Management, P.O. Box 177, 2600 MH Delft, Netherlands *Correspondence to:* E.J.C. Dupuits (e.j.c.dupuits@tudelft.nl)

[revised manuscript text omitted]

---

## Referee Report (RR1)

Updated Referee report on Dupuits et al., 2017, Using Graphs to find economically optimal safety targets for multiple lines of flood defences.

**General comment:**

The authors have thoroughly revised their paper. The paper now clearly discusses previous work on which this paper builds. The scientific value added of this paper is clear from the introduction section (section 1).

Unfortunately, the scientific value added of this paper is not properly described in the abstract. Rewriting this abstract will solve this problem. In addition, I have some technical and presentational remarks to obtain scientific correctness and improve the presentation.

The paper title should also be changed to correctly state the scientific value added. I propose as title: "Revisiting a graph based approach for solving economically optimal safety targets for flood defences to avoid annual expected damage calculations".

I propose to the editor that she handles these matters from now on.

Comments:

1.  Abstract: please state that: "This paper *revisits* an approach for …". Instead of ""This paper *presents* an approach….". As clearly described in the introduction, this paper revisits a solution approach earlier presented in previous papers and reports and implemented in one earlier paper.
2.  Abstract: delete the sentence: 'and is, thanks to some beneficial properties of the application able to traverse large problems'. This claim is not at all supported in the paper. As previously assessed and discusses in Zwaneveld and Verweij (2014a) and Eijgenraam et al. (2016 , including proceeding working paper), this shortest path approach is unable to solve present real world problems due to the fat that a shortest path approach requires exponential amount of combinations. This fact is also mentioned in this paper on page 6, line 15-20). Hence, the shortest path approach can be used for small problems only. Other solution approaches are better to handle large problems. That is the reason why previous authors did not use the shortest path based approach although they were fully aware of it. This must also be clearly discussed in the main text. My previous referee report included many references to that.
3.  Delete (and rephrase) the sentence: The work presented here make cost-benefit…. both easier and applicable to a broad range of flood defences with multiple lines of defences''. As follows from the detailed and cumbersome discussion of the shortest path implementation, I think this is not easy and broader applicable at all. I do think that the other approaches by Zwaneveld and Verweij (2014a) and Eijgenraam et al 92016) are much easier and more general applicable. See the discussion of pro and cons of several solution approaches (including shortest path) in Zwaneveld and Verweij (2014a) and Eijgenraam et al. (2016) I am sure that some readers prefer these other approaches. I am also sure that some readers prefer the shortest path approach. As discussion in section 1, this differs from once person to another. Hence, rephrase this sentence for example as: "The work presented here provide suggestions to implement the shortest path approach to cost benefit analyses of complex defence systems with interdependent multiple lines of defences."

4. Section 1: As described, the authors revisit the shortest path approach to reduce AED estimates. Other approaches (see page 4 line 4-8) can also be used or adjusted to avoid AED estimates. This should be mentioned. Be clear about the fact that shortest path approach may not be the best approach to avoid AED estimates. Leave the answer to this question for further research. I think that the heuristic approach and the ILP approach by Zwaneveld and Verweij (2014a) are first and second best approaches to find 'good solutions with minimal AED estimates'. Of course, this is an expert guess from me which I did investigate (yet). However, the authors didn't look into this question as well.

5. Section 2.2 and 2.3: the proposed shortest path solution approach is – in my opinion - a standard approach. I advise the editor not to publish these two subsections. I see no scientific value added. Please refer to textbooks and wikipedia internetpages instead. This will make the paper shorter. Furthermore, it avoids several mistakes. The authors seem to be non –experts in shortest path algorithms. E.g.:

   a. Many scientists consider Dykstra algorithm and UCS as logically identical[1].
   b. A greedy shortest path algorithm is – in general - not a greedy algorithm. The first provide a proven optimal solution. The latter provide a quick heuristical (i.e. possibly non-optimal) solution.
   c. The presented shortest path algorithm gives – by its well-known structure- a proven optimal solution. The authors do not seem to be aware of this.
   d. Page 9, line 4. Should $t$ not be 200? Other similar mistakes in line 5.

6. I do not see the scientific value added of section 3.1. My advice is to delete this section.

7. Figure 17: please refer that this approach describes the heuristic approach as previously proposed by Zwaneveld and Verweij (2014a). See page 4, line 5 in which this approach is already mentioned by the authors.

8. Section 4.1: new title: "Single flood defence with tiny step sizes".

9. Section 4.1:  Eijgenraam 2006 presents an analytical solution which may be non-optimal. Eijgenraam et al. (2016) presents an analytical, proven optimal solution. Please use the latter. If you stick to Eijgenraam (2006): please clearly state that the solution may be non-optimal!

10. Section 4.2: New title : "Single flood defence with regular step sizes"

11. Section 5: The authors should state explicit here that they *revisit* the shortest path approach to avoid AED estimates.

12. Section 6: please rewrite this section as suggested for the abstract.
* * *
[1] See for example. Felner, 2011 Position Paper: Dijkstra's Algorithm versus Uniform Cost Search or a Case Against Dijkstra's Algorithm. Proceedings, The 4th SoCS 2011.

---

## Author Response (AR2)

**Reviewer #1**

To me the main criteria for acceptance is how well the authors have addressed the other reviewers concerns regarding novelty and potential plagiarism. I will leave it to that reviewer to assess that issue.

Thank you for this comment. We have addressed these issues in our previous reply to the other reviewer.

My main concern was that the paper appeared rather messy with a poor method description and an amount of simulations similar to what is done in traditional CBA. It seems that the authors have responded to the latter by only comparing to LP, which undoubtedly contains more simulations than the proposed method.

Thank you for this comment. We indeed meant to compare the number of simulations with a similar implementation (which uses LP and a similar holistic view of flood defences).

The definition of risk has been improved substantially, but there are still errors and inconsistencies. In particular, the authors throughout the paper discusses how AED (normally denoted EAD, Expected Annual Damage) is affected. But the term is never defined. Equation (1) defines TC but the relationship between TC and EAD is never stated.

Thank you for this comment, we have further improved the following:

- We changed AED into EAD
- We had a link between R and EAD in text, which we now added to Eq2 as well.

It would also be nice to define Eq (1) properly. I assume that you make the summaztion over forecast horizon, but Figure 1 could indicate something else. Also, the choice of forecast horizon is not trivial when you make projections over several centuries.

Thank you for this comment:

- We added further details to Eq1, specifically we added the forecast horizon to the summations and explicitly show that both the EAD and the investment costs are time dependent.
- Choice of forecast horizon is indeed non-trivial, as (for example) the uncertainty of variables used within the economic optimisation can depend on this choice. For this reason, we left the forecast horizon as configurable parameter. Furthermore, as we briefly touch upon in the examples section, a finite forecast horizon will somewhat influence the economic optimisation itself. Specifically, decisions right before the end of the forecast horizon. However, as this paper focuses mostly on an approach for economic optimisation and its implementation, which builds upon previous similar approaches, we think that a discussion regarding forecast horizon is not necessary in this paper.

Further, if Eq (2) should be correct D\_flood should be defined as the expected loss incurred due to flooding, not the annual expected loss incurred due to flooding. Thank you for this comment. We have corrected the description per your suggestion.

Another issue related to the scoping of the paper is Fig 3 and corresponding discussion on page 3. There the main driver for the paper seems to be a more holistic view of flood

defences, because several defences might impact one another. In Fig 3 the main argument seems to be due to risk of dike breaches, but this is never touched upon again later in the paper.

Thank you for this comment. You are right that a more holistic view of flood defences is one of the drivers for writing this paper, though the direct reason for writing this paper is to deal with the increase of the number of computational costly risk cost calculations because of the more holistic view of flood defences. Regarding Fig 3:

• We have slightly altered Fig 3 and its description

• We have added a more explicit description and reference to Fig 3 in the main text Furthermore, we have replaced 'multiple lines of defence' with 'interdependent flood defences', as the latter seems a more fitting, generic description of the type of flood defence systems we discuss in the paper.

So while I still see some value of visualizing a structured approach to identification of (future) investments I still cannot recommend publication in its current form. Language should also be improved, in particular I would recommend a consistant use of tense. I have not read the following chapters in detail but will do so if the authors get a third chance and have responded to the above comments.

Thank you for this comment. We have revised the language and use of tense, particularly regarding the abstract, introduction, discussion and conclusions.

In doubt about whether this is minor or major revision. I have ticked major because they have already been asked to improve the above issues once.

**Reviewer #2**

**General comment:**

The authors have thoroughly revised their paper. The paper now clearly discusses previous work on which this paper builds. The scientific value added of this paper is clear from the introduction section (section 1).

Unfortunately, the scientific value added of this paper is not properly described in the abstract. Rewriting this abstract will solve this problem. In addition, I have some technical and presentational remarks to obtain scientific correctness and improve the presentation. Thank you for these comments. As these comments seem to be repeated in more detail in the text below, we will answer these comments in the following text.

The paper title should also be changed to correctly state the scientific value added. I propose as title: "Revisiting a graph based approach for solving economically optimal safety targets for flood defences to avoid annual expected damage calculations".

Thank you for this comment. We have rephrased the title to be more precise: "Economically optimal safety targets for interdependent flood defences in a graph-based approach with an efficient evaluation of expected annual damage estimates". Regarding the word 'revisiting', see also comments 1, 11 and 12.

I propose to the editor that she handles these matters from now on.

**Comments:**

1. Abstract: please state that: "This paper **revisits** an approach for ...". Instead of ""This paper **presents** an approach....". As clearly described in the introduction, this paper revisits a solution approach earlier presented in previous papers and reports and implemented in one earlier paper.

Thank you for this comment. We replaced the word 'presents' with the word 'advances', as we think this word more accurately covers the intent and content as opposed to 'revisits': we build upon existing approaches.

2. Abstract: delete the sentence: 'and is, thanks to some beneficial properties of the application able to traverse large problems'. This claim is not at all supported in the paper. As previously assessed and discusses in Zwaneveld and Verweij (2014a) and Eijgenraam et al. (2016, including proceeding working paper), this shortest path approach is unable to solve present real world problems due to the fat that a shortest path approach requires exponential amount of combinations. This fact is also mentioned in this paper on page 6, line 15-20). Hence, the shortest path approach can be used for small problems only. Other solution approaches are better to handle large problems. That is the reason why previous authors did not use the shortest path based approach although they were fully aware of it. This must also be clearly discussed in the main text. My previous referee report included many references to that.

Thank you for this comment. We meant to refer to our particular implementation which is able to represent large graphs in a memory efficient manner. The above

comment seems to refer to solving flood defence systems with a large amount of (partial) independent flood defences, for which the shortest path approach is not the most efficient solution. We also want to note that we gave extensive replies to similar comments in the previous review report. We have changed the relevant sentence in the abstract to "and is, thanks to some beneficial properties of the application, able to represent large graphs with strongly reduced memory requirements." This rephrasing should make it clear that we are referring to the implementation of large graphs for the application of interdependent flood defence systems, not the size of the flood defence system. The motivation for this statement can be found in Section 3.1.

3. Delete (and rephrase) the sentence: The work presented here make cost-benefit.... both easier and applicable to a broad range of flood defences with multiple lines of defences". As follows from the detailed and cumbersome discussion of the shortest path implementation, I think this is not easy and broader applicable at all. I do think that the other approaches by Zwaneveld and Verweij (2014a) and Eijgenraam et al 92016) are much easier and more general applicable. See the discussion of pro and cons of several solution approaches (including shortest path) in Zwaneveld and Verweij (2014a) and Eijgenraam et al. (2016) I am sure that some readers prefer these other approaches. I am also sure that some readers prefer the shortest path approach. As discussion in section 1, this differs from once person to another. Hence, rephrase this sentence for example as: "The work presented here provide suggestions to implement the shortest path approach to cost benefit analyses of complex defence systems with interdependent multiple lines of defences."

Thank you for this comment. We agree with the part that everyone may have their own preference. However, the discussion we present for the shortest path implementation is specific for systems of interdependent flood defences, and how this particular implementation can be done in a generic way. Therefore, we have changed the sentence into: "The proposed approach is set up in a generic way and implements the shortest-path approach for optimising cost-benefit analyses of interdependent flood defences with computationally expensive flood risk calculations."

4. Section 1: As described, the authors revisit the shortest path approach to reduce AED estimates. Other approaches (see page 4 line 4-8) can also be used or adjusted to avoid AED estimates. This should be mentioned. Be clear about the fact that shortest path approach may not be the best approach to avoid AED estimates. Leave the answer to this question for further research. I think that the heuristic approach and the ILP approach by Zwaneveld and Verweij (2014a) are first and second best approaches to find 'good solutions with minimal AED estimates'. Of course, this is an expert guess from me which I did investigate (yet). However, the authors didn't look into this question as well.

Thank you for this comment.

a) We improved the description in section 1 by adding the following line right before the aim of the paper: "One such optimisation routine that can be easily implemented in a general programming language and adapted to use lazy evaluation is the shortest-path approach". We think this sentence indicates that our choice is just one of the choices that can be made.

- b) We indeed did not look into all the possible solution strategies, and can therefore not state with certainty whether all these possible strategies are better or even valid candidates. Therefore, we refrain from making any qualification regarding which other strategies would be applicable or better.
- 5. Section 2.2 and 2.3: the proposed shortest path solution approach is in my opinion a standard approach. I advise the editor not to publish these two subsections. I see no scientific value added. Please refer to textbooks and wikipedia internetpages instead. This will make the paper shorter.

Thank you for this comment. We have not removed Section 2.2 and 2.3, for reasons listed in our reply to 5a.

*Furthermore, it avoids several mistakes. The authors seem to be non –experts in shortest path algorithms. E.g.:*

- a) Many scientists consider Dykstra algorithm and UCS as logically identical1. Thank you for this comment. As you already mentioned (using a reference we already use in our paper), Dykstra is indeed logically equivalent to UCS. However, the main message of that same reference is that the commonly used implementations for Dykstra and UCS are not equivalent. We specifically use the UCS implementation. The UCS implementation is the fundamental basis for reducing the number of required EAD calculations (which is explicitly mentioned and used in section 3.3). Therefore, removing section 2.2 and 2.3 would make it hard to explain how, why and where the reduction of EAD calculations takes place. Furthermore, as mentioned in our reply to the previous review report, we think the target audience of this journal might not all be experts in operations research. Because of these two reasons, we have not deleted Section 2.2 and 2.3.
- b) A greedy shortest path algorithm is in general not a greedy algorithm. The first provide a proven optimal solution. The latter provide a quick heuristical (i.e. possibly non-optimal) solution.

Thank you for this comment. Your description differs from the one we use. Our description of greedy shortest path algorithms (and dynamic programming as well) follows from the book 'Introduction to algorithms' by T.H. Cormen (2009). See for example the chapters 15 (Dynamic Programming) and 16 (Greedy algorithms). Specifically, see:

- Chapter 24, page 644: "Dijkstra's algorithm, which we shall see in Section 24.3, is a greedy algorithm..."
- Chapter 24, page 659: "Because Dijkstra's algorithm always chooses the "lightest" or "closest" vertex in V-S to add to set S, we say that it uses a greedy strategy."
- c) The presented shortest path algorithm gives by its well-known structure- a proven optimal solution. The authors do not seem to be aware of this. Thank you for this comment. We are aware: see section 2.4 where we

explicitly mention this, with references. We mention this again in the last paragraph of the conclusion.

- d) Page 9, line 4. Should t not be 200? Other similar mistakes in line 5.
   Thank you for this comment. This was a typo, we mentioned vertex 22 where we should have mentioned vertex 12. We have corrected the relevant text.
- 6. I do not see the scientific value added of section 3.1. My advice is to delete this section.

Thank you for this comment. Section 3.1 is relevant regarding the specifics of our implementation of the shortest-path implementation. (This is also directly related to comment 3 and our answer to that comment). Section 3.1 describes the repetitiveness in the adjacency lists of the vertices in a graph for the economic optimisation of interdependent flood defences. Acknowledging this repetitiveness results in needing only a single adjacency list per graph, instead of an adjacency list per vertex. This strongly reduces the memory requirements for our specific implementation, compared to generic versions of the shortest-path implementation. Therefore, we disagree with the advice to delete this section.

7. Figure 17: please refer that this approach describes the heuristic approach as previously proposed by Zwaneveld and Verweij (2014a). See page 4, line 5 in which this approach is already mentioned by the authors.

Thank you for this comment.

- *a*) We indeed briefly mention the possible solution strategies of Zwaneveld and Verweij (2014) in the introduction as these strategies are mentioned in the referred citation.
- b) We have again reviewed the mentioned citation and could only find a brief reference to the heuristic approach mentioned by reviewer #2. This brief reference cites an appendix of another document which is not available online. We have a copy of this appendix (which is classified as an internal memo and is in Dutch), which again briefly mentions a heuristic solution in another memo. We do not have this memo and could not find (in the short period between reviews) the latter memo. As we cannot verify the details of the mentioned heuristic, we cannot assess the degree of similarity.
- c) Furthermore, in the text relevant to Figure 17, we only mention that this is a possible way to more efficiently represent the graph for a system with interdependent and independent flood defences. No mention is made of a possible way to solve this graph, and we explicitly mention that we did not implement this graph representation. A heuristic would be one step further (i.e. a way to solve this). As we do not mention any solution for the

conceptual graph representation, we do not see how we present a heuristic. At this point, we have to opt against directly citing a reference we cannot verify ourselves, and which seems to be an internal memo in Dutch which would be hard to validate for international readers. Furthermore, because we don't describe an actual solving strategy we do not see how we describe a heuristic.

**8. Section 4.1: new title: "Single flood defence with tiny step sizes". Thank you for this comment. We think we made it clear in the text why we used these small steps sizes (to obtain an accurate comparison with the existing analytical answer). Therefore, we opt against rephrasing the section title.**

9. Section 4.1: Eijgenraam 2006 presents an analytical solution which may be nonoptimal. Eijgenraam et al. (2016) presents an analytical, proven optimal solution. Please use the latter. If you stick to Eijgenraam (2006): please clearly state that the solution may be non-optimal!

Thank you for this comment. We used the data listed in Eijgenraam (2006), but we already used the analytical solution of Eijgenraam (2016) to calculate the results. We have added the following sentence to make this clear: "The numerical results were re-calculated with the solution listed in Eijgenraam et al. (2016)..."

- 10. Section 4.2: New title : "Single flood defence with regular step sizes" Thank you for this comment. In section 4.2, we look at two independent (identical) flood defences. Therefore, we opt against rephrasing the section title.
- 11. Section 5: The authors should state explicit here that they revisit the shortest path approach to avoid AED estimates.

Thank you for this comment. We did add the word 'advances' to the conclusions in section 6 (similar modification as in the abstract, see also comment 1). The requested change is now mentioned in the abstract, introduction and conclusions: we think this is sufficient and it does not need to repeated again in the discussion.

**12. Section 6: please rewrite this section as suggested for the abstract.**

Thank you for this comment. Per our replies to comments 1,2 and 3 we think we have made the existing claims in the conclusions sufficiently clear. The requested change in comment 1 is discussed in the previous comment (11), while our replies to comments 2 and 3 resulted in better and more explicit communication of our contributions in the abstract. These contributions were already present in the conclusions. We only made some minor changes to make the conclusions consistent with the changes to the abstract.

**Using graphs to find economically Economically optimal safety targets for multiple lines of interdependent flood defences in a graph-based approach with an efficient evaluation of expected annual damage estimates**

Egidius Johanna Cassianus Dupuits1, Ferdinand Lennaert Machiel Diermanse2, and Matthijs Kok1 1Delft University of Technology, Faculty of Civil Engineering and Geosciences, P.O. Box 5048, 2600 GA Delft, Netherlands 2Deltares Unit Inland Water Systems, department of Flood Risk Management, P.O. Box 177, 2600 MH Delft, Netherlands *Correspondence to:* E.J.C. Dupuits (e.j.c.dupuits@tudelft.nl)

Abstract. Flood defences can be designed as multiple lines of defencedefence systems can be seen as multiple interdependent flood defences. This paper presents advances an approach for finding an optimal configuration for flood defence systems, based on an economic cost-benefit analysis with an arbitrary number of interdependent lines of defenceflood defences. The proposed approach is based on a graph algorithm and is, thanks to some beneficial properties of the application, able to traverse

- 5 large problems represent large graphs with strongly reduced memory requirements. Furthermore, computational efficiency is achieved by delaying cost calculations until they are actually needed by the graph algorithm. A number of case studies were carried out This significantly reduces the required number of computationally expensive flood risk calculations. In this paper, we conduct a number of case studies to compare the optimal paths found by the proposed approach with the results of competing methods , and were found to that generate identical results. The work presented here makes proposed approach is set up in a
- 10 generic way and implements the shortest-path approach for optimising cost-benefit analyses of complex flood defence systems with interdependent multiple lines of defence both easier and applicable to a broad range of flood defence systems with multiple lines of defence interdependent flood defences with computationally expensive flood risk calculations.

**1 Introduction**

Concerns regarding the safety of people and assets in flood prone areas has led to the construction of flood defence systems all around the world. Some flood prone areas, for example a large part of the Netherlands, face huge potential loss of life and economic value in case heavy flooding occurs. This has led to extensive research regarding estimating the the estimation of flood risk in flood risk of flood prone areas. Coupled to this quantification of the flood risk, is the question of 'how safe' a flood prone area should be and what the acceptable risk should be (Vrijling et al., 1998). An often used approach to help answer this question is a cost-benefit analysis.

Economic optimisation of flood defences, as applied in the Netherlands, is based on a cost-benefit analysis of the <del>flood risk</del> <del>cost reduction sum of the annual flood risks</del> balanced against the sum of the investment costs for flood defences. This type of cost-benefit analysis was <del>already</del> originally developed in the 1950's by Van Dantzig (1956), and is still used and discussed to

this day (Eijgenraam, 2006; Kind, 2014). The basic principle behind the economic optimisation of flood defences is finding the minimum of the total costs and is as illustrated in Figure 1. The total costs (TC, Eq. 1) are the sum of the annual risk costs ( $\sum R \sum_{t=0}^{p} R(t)$ ) and investment costs ( $\sum I$ ) for  $\sum_{t=0}^{p} I(t)$ ) over a given time period. An annual risk cost (R) is pyears). The total costs are expressed as the present value of the (future) annual risk costs and investment costs, which means

5 these costs are discounted at a discount rate r. The annual risk cost R(t) is defined in Eq. 2 defined as the annual probability of flooding  $(P_{flood})$ times the annual expected loss incurred at time t  $(P_{flood,t})$ , multiplied by the expected damages due to flooding  $(D_{flood}$  at time t  $(D_{flood,t})$ . An alternative term for the annual risk cost is the Annual Expected Expected Annual Damage, or AEDEAD. Generally speaking, a larger investment will lead to a lower AED and EAD; this is where the economic optimisation tries to find an optimal solution (i.e. the lowest total cost).

10
$$TC = \sum_{t=0}^{p} R(t)e^{-rt} + \sum_{t=0}^{p} I(t)e^{-rt}$$
(1)

(2)

$$R(t) = \text{EAD}(t) = P_{flood flood, t} \cdot D_{flood flood, t}$$

Safety level  $\rightarrow$

[revised manuscript text omitted]